

# Soil type and fertilizer rate affect wheat (*Triticum aestivum* L.) yield, quality and nutrient use efficiency in Ayiba, northern Ethiopia

Weldemariam Seifu[1,2], Eyasu Elias[2], Girmay Gebresamuel[3] and Wolde Tefera[4]

[1] Department of Horticulture, Salale University, Fiche, Oromia, Ethiopia
[2] Center for Environmental Sciences, Addis Ababa University, Addis Ababa, Ethiopia
[3] Department of Land Resources Management and Environmental Protection, Mekelle University, Mekelle, Tigray, Ethiopia
[4] Department of Plant Science, Salale University, Fiche, Oromia, Ethiopia

Corresponding author
Weldemariam Seifu,
weldemariam.seifu@aau.edu.et

## ABSTRACT

The blanket NP fertilizer recommendation over the past five decades in Ethiopia did not result in a significant increment of crop productivity. The main lack of success was highly linked to the extrapolating approach of one site success to others without considering the climate, soil, and ecological setting and variations. As a result, a new fertilization approach was desperately needed, and with this premise, new blended fertilizers are now being introduced to replace the conventional approach. Thus, the objective of this study was to examine the effect of NPSZnB blended fertilizer on bread wheat yield attributes, quality traits and use efficiency in two different soil types under rain-fed conditions in Ayiba, northern Ethiopia. Relevant agronomic data were evaluated and recorded from plots of each soil types for analysis. The analysis of variance revealed a significant ($p < 0.001$) variation on all the agronomic and grain quality traits due to the main and interaction effects of soil type and fertilizer treatment factors. Most agronomic and quality characteristics recorded the highest result in the highest treatment applications (175 and 150 kg NPSZnB ha$^{-1}$) in both soils. Yield and grain quality traits of bread wheat was also found better under fertilized plots than unfertilized plots. In both soil types increasing application of the new blended fertilizer rate from 50–175 kg NPSZnB ha$^{-1}$ showed an increasing trend in grain yield from 1.6 to 4.3 and 2.5 to 5.4 t ha$^{-1}$ in Vertisol and Cambisol soils, respectively. The varied yield as a response of fertilizer treatments across soils signifies soil-specific fertilization approach is critically important for production increment. On the other hand, based on the partial budget analysis the highest net benefit with the highest marginal rate of return in both Vertisol and Cambisol soils were obtained when treated with 100 and 125 kg NPSZnB ha$^{-1}$, respectively. Therefore, to produce optimum bread wheat yield under rainfed conditions in Ayiba (northern Ethiopia) fertilizing Vertisols with 100 kg NPSZnB ha$^{-1}$ and fertilizing Cambisols with 125 kg NPSZnB ha$^{-1}$ is recommended.

# INTRODUCTION

Ethiopia is one of the few nations in Sub-Saharan Africa (SSA) where agriculture is the backbone of the economy (*Baye, 2017*; *Elias, 2016*; *Plecher, 2019*), with crop production accounting for around 28% of GDP (*NBE National Bank of Ethiopia, 2019*). Within the agriculture sector, cereals are the principal staple crops in terms of planted area (81.46%) and output volume (88.52%). Wheat (*Triticum aestivum*), along with teff (*Eragrostis teff* Zucc), maize (*Zea mays*), sorghum (*Sorghum bicolor*), and barley (*Hordeum vulgare*), is one of the key grains that are at the heart of Ethiopia's agricultural and food economy (*Bishaw, 2004*; *Tesfaye, Laekemariam & Habte, 2021*). Wheat represents 13.91% of the total cereal planted area and 81% of the total food output volume (*CSA, 2020*). Our world produces food for over 10 billion people; however, 815 million still suffer from hunger or malnutrition (*FAO, IFAD, UNICEF, WFP, WHO, 2017*; *Holt-Giménez et al., 2012*). Likewise, Ethiopia produces the most wheat in SSA, although it is still a net importer (*Hodson et al., 2020*). For the past five decades, research in Ethiopia has been focused on fertilizer recommendations of urea (46% N) and DAP (18% N-46% $P_2O_5$) as a blanket application considering the only limiting nutrient of Ethiopian soils through various agricultural extension efforts (*Alemu et al., 2016*; *Biratu, 2008*; *Desta & Almayehu, 2020*; *Elias, Okoth & Smaling, 2019*; *Fisseha et al., 2020*).

However, the present national average wheat yield (2.97 t ha$^{-1}$) is much lower than the potential yield (Yp = 9.6 t ha$^{-1}$), water-limited potential yield (Yw = 8.3 t ha$^{-1}$) (http://www.yieldgap.org/Ethiopia), and world average yield (4.39 t ha$^{-1}$) (*Purdy & Langemeier, 2018*). As a result, millions of households endure chronic food insecurity each year, and their survival is reliant on humanitarian food aid (*Elias, Okoth & Smaling, 2019*; *Elias & Van Beek, 2015*). Moreover, currently, the increase in population and the modification of food habits connected to urbanization are inflicting to surpass the demand for national wheat supply (6.3 Mt of demand against 4.6 Mt of supply) in Ethiopia. As a result of the paucity of supply, the current output is insufficient to meet domestic needs, forcing the country to import up to 50% of its needs from the Black Sea region in recent years (*Elias, Okoth & Smaling, 2019*; *Hodson et al., 2020*; *Minot et al., 2019*). Although demand and output are not yet linked, good extension support and the use of suitable inputs have the potential to boost wheat yield (*Habte et al., 2020*; *MoA, 2019*). As a result, the Ethiopian government recently announced that it will close the yield gap (*van Ittersum et al., 2013*) and reduce wheat imports from 1.7 million metric tons in 2019 to zero in 2023, allowing the country to become self-sufficient through acid and Vertisol soil management, intensification, extensive irrigation, and agricultural mechanization in collaboration with the private sector (*Getachew, 2020*; *MoA, 2019*; *Simret, 2019*). Apart from that, to achieve this aim, it will be necessary to apply balanced amounts of the most critical nutrients to increase yield while decreasing nutrient losses; that is when fertilization

is fine-tuned to local soil chemical conditions and crop requirements (*Elias, Okoth & Smaling, 2019*; *Roy et al., 2006*). Therefore, accelerating production in Africa is critical to achieving self-sufficiency or at least to sustaining the existing self-sufficiency ratio as the massive population increase occurs by 2050 (*Elrys et al., 2021*; *Elrys et al., 2020*).

The fundamental cause of low yield in Africa's small farming system is soil nutrient depletion and inefficient mineral fertilizer application (*Elrys et al., 2019*; *Hailu et al., 2015*; *Harfe, 2017*). The rate of soil macronutrient depletion in Africa (Ethiopia) *per annum* was estimated at 22 (122) kg N ha$^{-1}$, 2.5 (13) kg P ha$^{-1}$, and 15 (82) kg K ha$^{-1}$, respectively (*Haileslassie et al., 2005*; *Sanchez, 2002*). However, in the wheat-growing portions of Ethiopia, production stalling reasons include pest problems (diseases, weeds, and insects), erratic rainfall incidence, and others (*Abera & Kassa, 2017*; *Elias, Okoth & Smaling, 2019*; *Harfe, 2017*; *Mulugeta et al., 2017*; *Walsh et al., 2020*). The fertilization problem revealed the need for having a high possibility for including lacking critical minerals in fertilization techniques. Micronutrients including B, Fe, Mn, Zn, and Cu are vital for plant health and growth, despite their little levels (*Waqeel & Khan, 2022*). Hence, by considering all essential plant nutrients in fertilizer sources and fertilization strategies, it is possible to increase nutrient use efficiency and possibly intensify yield (*Bindraban et al., 2015*; *Dimkpa & Bindraban, 2016*). Currently, soil scientists have noted that the problem of undernourishment begins with not feeding the soil, which continues to devastate many populations due to the 'hidden hunger' of essential minerals and vitamins (*Shekhar, 2013*; *von Grebmer et al., 2014*), which is very common and chronic in SSA countries due to socioeconomic and geospatial reasons (*Gashu et al., 2021*). Because of the complex interplay between the local environment and locally relevant crops, soil testing and, more importantly, soil-test-based recommendations must be site-specific (*Kedir, Zhang & Unc, 2021*).

The previously described blanket recommendation usually fails to account for variances in resource endowment (soil type, labor capacity, climatic risk) or allows for substantial fluctuations in the input/output price ratio, deterring farmers from using fertilizer. Similarly, research has shown that the blanket reference nutrients are not agronomically balanced and that their ongoing usage is diminishing soil nutrient supplies (*Elias, 2016*; *Elias & Van Beek, 2015*; *Tewolde, Gebreyohannes & Abrha, 2020*). According to other studies, different plant species respond differently to fertilizer rates and combinations in different soil types, and balanced fertilizers are necessary to enhance growth, yield, quality, and efficiency (*Akamine et al., 2007*; *Chowdhury et al., 2008*). As a result, soil test-based fertilizer use, particularly those combined with S, B, Zn, and other nutrients, is advised rather than a blanket recommendation in avoiding difficulties caused by nutrient-deficient soil (*ATA, 2016*). In trials from throughout the country, major grain crops reacted well to mixed fertilizer application rather than NP alone (*Desta & Almayehu, 2020*; *Fayera, Adugna & Mohammed, 2014*; *Li et al., 2019*; *Tesfay & Gebresamuel, 2016*), although the absence of one or more nutrients besides NP can dramatically impair production. As a result, Ethiopia is undertaking a strategy to use diverse blended fertilizer sources to minimize productivity limits and become self-sufficient in agricultural production (*ATA, 2016*). The Emba Alaje area (Fig. S1), which includes our research site,

Ayiba, requires five types of mixed fertilizers (Fig. S1A), plus potash fertilizer (Fig. S1B), according to *EthioSIS (2017)*. The extensive coverage of NPSZnB blend fertilizer among the five suggested lists (Fig. S1A) indicated that this compound fertilizer is essential to promote production. As a result, the purpose of the site-soil-specific experiment is to create fertilizer recommendations relevant to each microclimate location's soil type and natural fertility condition (*Elias, 2018*). Finally, site-specific management improves profitability by increasing agricultural productivity while also safeguarding the environment (*Teklu & Michael, 2007*).

Balanced fertilization is necessary to ensure crop yield. In Ethiopia, researchers are looking at the effects of novel mixed fertilizer sources on diverse crops produced in varied soil types. Despite the publication of a soil fertility atlas and the development of a new blended fertilization method, information on site-specific mixed fertilizer rates for various crops remains scarce. To our knowledge, the effect of varied NPSZnB mixed fertilizer rates on bread wheat production in different soil types in the Ayiba highland, which we employed as a case study, has not yet been investigated. As a result, a rainfed field experiment was carried out to answer the following questions: (i) how does the selected test crop respond to increasing rates of NPSZnB blended fertilizer in Vertisol and Cambisol soil types? and (ii) what is the comparative advantage of NPSZnB blended fertilizer over conventional NP recommendation in both soils. Therefore, the following objectives are being assessed in this study: (a) to determine and recommend the optimum rate of NPSZnB blended rate for wheat production on Ayiba's Vertisol and Cambisol soil types under rainfed conditions, and (b) to investigate how soil type affects bread wheat yield and grain quality response to NPSZnB blended fertilizer application.

## MATERIALS AND METHODS

### Site description: location, climate, soil, and husbandry

The research was carried out in farmers' fields in the Ayiba watershed (4,099.14 ha) of the Emba-Alaje district in southern Tigray, northern Ethiopia. The area is located between 12°51′18′′–12°54′36′′N and 39°29′24′′–39°35′24′′E. The elevation varies between 2,722 and 3,944 m above sea level. The area is one of Tigray's potential wheat-producing regions, with a tepid to cool semi-arid highland agro-ecological zone (*Amanuel, Girmay & Atkilt, 2015*; *Elias, 2016*; *Negash & Israel, 2017*). The landform of the study area is dominated by high mountainous relief hills and starkly dissected plateaus with steep slopes (>30% slope gradient) complemented by valley bottoms and river gorges (*Amanuel, Girmay & Atkilt, 2015*; *Elias, 2016*). Based on the long-term meteorological data rainfall in the study area has generally bimodal characteristics where the main Keremti (summer: June to September) season is preceded by a small rainy season called Belgi (spring: February to May) predominantly derived from the Indian Ocean (*Yemane, Weldemariam & Girmay, 2020*). The total amounts of rainfall received during the 2017 and 2018 cropping seasons were 417 and 479 mm, respectively (*Mesfin et al., 2020*). The mean minimum and maximum temperatures were 12.6 and 23.3 °C for the 2017 cropping season and 11.6 and 22.3 °C for the 2018 cropping season, respectively (*Mesfin et al., 2020*). The area's annual potential evapotranspiration (PET) is about 1,411 mm (*Elias, 2016*).

**Table 1 Information on on-farm experimental fields.**

| Farms | Geographical location | | | Crop history | Soil parent material[1,2] | Soil type |
|---|---|---|---|---|---|---|
| | Latitude (N) | Longitude (E) | Altitude (m) | | | |
| Farm - 1 | 12°52′28.1″ | 39°32′38.1″ | 2,219 | Teff | Alluvial | Haplic Vertisol |
| Farm - 2 | 12°54′04.5″ | 39°32′16.1″ | 2,468 | Field bean | Fluvial | Haplic Vertisol |
| Farm - 3 | 12°53′49.1″ | 39°32′56.6″ | 2,744 | pea | Fluvial | Haplic Cambisol |
| Farm - 4 | 12°53′54.6″ | 39°31′23.3″ | 2,468 | Teff | Alluvial | Haplic Cambisol |

Notes:
[1] *Elias (2016)*.
[2] *Amanuel, Girmay & Atkilt (2015)*.

**Table 2 The nutrient share of each element in each treatment.**

| Code | Treatments (kg ha$^{-1}$) | Nutrient composition | | | | | | TNA |
|---|---|---|---|---|---|---|---|---|
| | | N (total) | P$_2$O$_5$ | K$_2$O | S | Zn | B | |
| T$_1$ | Control (no fertilizer) | 0 | 0 | 0 | 0 | 0 | 0 | 0 |
| T$_2$ | 100 DAP+50 KCl+100 Urea | 64 | 46 | 25 | 0 | 0 | 0 | 135 |
| T$_3$ | 50 NPSZnB+50 KCl+119.8 Urea | 64 | 17.9 | 25 | 3.9 | 1.1 | 0.05 | 111.95 |
| T$_4$ | 75 NPSZnB+50 KCl+110 Urea | 64 | 26.8 | 25 | 5.8 | 1.7 | 0.08 | 123.38 |
| T$_5$ | 100 NPSZnB+50 KCl+100.4 Urea | 64 | 35.7 | 25 | 7.7 | 2.2 | 0.1 | 134.70 |
| T$_6$ | 125 NPSZnB+50 KCl+90.7 Urea | 64 | 44.6 | 25 | 9.6 | 2.8 | 0.13 | 146.13 |
| T$_7$ | 150 NPSZnB+50 KCl+81.1 Urea | 64 | 53.6 | 25 | 11.6 | 3.3 | 0.15 | 157.65 |
| T$_8$ | 175 NPSZnB+50 KCl+71.3 Urea | 64 | 62.5 | 25 | 13.5 | 3.9 | 0.18 | 169.08 |

Note:
Compiled Based on *NPSZnB* blend = 17.8 N - 35.7 P$_2$O$_5$ - 7.7 S - 0.1 B -2.2 Zn, Urea = 46-0-0, and DAP = 18-46-0. Urea was added for all *NPSZnB* treatment to adjust to its recommended rate of 64 kg ha$^{-1}$, TNA, total nutrient applied per treatment.

Volcanic trap-rocks are common parent materials in the study area, with a primary basalt lithology, on which Vertisols, Cambisols, Regosols, and Leptosols have developed (*Amanuel, Girmay & Atkilt, 2015*; *Elias, 2016*), with dominant clay texture derived primarily from fluvial and alluvial sediments (Table 1). Mixed farming is the most common farming system. (*Seifu et al., 2021*). Cereal and legume crops and some vegetable and fruit crops are grown in the study area (*Girmay et al., 2014*). Natural pasture is the primary source of animal feed in areas where farmers practice intensive pasture land grazing with a higher stocking rate, resulting in poor natural pastureland management (*Atsbha et al., 2020*).

## Experimental set up: treatments, design, and parameters evaluated

To develop an optimal NPSZnB blended fertilizer application rate for bread wheat productivity, field experiments were conducted in two different soil types, namely Vertisol and Cambisol (Fig. S2). The experimental treatment included eight levels of fertilizer: control or no fertilize (T$_1$), urea + DAP each 100 kg ha$^{-1}$ (T$_2$) and six NPSZnB blended fertilizer rates - 50, 75, 100, 125, 150, and 175 kg ha$^{-1}$ (T$_{3-8}$) (Table 2). This field experiment included a blanket NP fertilizer application as a positive control for comparison. The new NPSZnB blended fertilizer is tested in this study with intention to

replace the traditional experience. Each block had eight plots measuring $3 \times 3$ m, with a distance of 1 m between blocks and 0.5 m between plots. The experimental fields were kept to 15 rows, each spaced by 0.2 m (*CIMMYT, 2013*). All agronomic practices were kept consistent across all treatments following the specific recommendation for bread wheat cultivation (*EIAR, 2007*). As a base fertilizer, whole doses of NPSZnB, DAP, and KCl were applied before sowing. Simultaneously, urea was applied in a split method, with 60% used as a basal dressing at sowing. The remaining 40% was applied as a top-dressing one month after sowing and after weeding in ideal moisture conditions or after rain. Wheat seed was applied at a rate of 125 kg ha$^{-1}$, and sowing was accomplished with a hand drill.

As a test crop, the bread wheat variety King-bird (ETBW 8512) was used. The variety was chosen for its early maturity and yield performance, high bread-making quality, multi-disease resistance, and adaptability to a wide range of agro-ecological conditions (low to mid-altitude) (*BGRI, 2015*; *CIMMYT, 2015*). Seed and fertilizer were purchased from the Ayiba Kebele Farmers' Cooperative Union (the smallest administrative unit in Ethiopia). Finally, at physiological maturity the yield and quality parameters listed below were assessed: Total tillers (TT) and productive tiller (PT), number of kernels per spike (NKS), thousand kernel weight (TKW), grain yield (GY), straw yield (SY), Biological yield (BY), harvest index (HI), grain protein content (GPC), and hectoliter weight (HLW). The wheat yield and yield component parameters were calculated using the national standard (*Abera et al., 2020a*), while the GPC and HLW parameters were calculated using AACC methods 46-11.02 and 55.10.01, respectively (*AACC International, 2002*).

## Soil sampling and analysis

Before starting the experiments, soil samples from topsoil (0–30 cm) were collected from the research farms using the grid sampling approach. They were bulked together, air-dried, then sieved using a 2-mm sieve to establish their physical and chemical properties. An auger was used to gather 18 soil samples (9 augers $\times$ 2 soil types) from the experimental blocks of each soil type. Each soil type's samples were carefully mixed. Each soil type received a kilogram of the composited sample *via* the quartering technique for laboratory examination (*Abera et al., 2020a*).

A hydrometer was used to determine soil texture (*Bouyoucos, 1962*). Textural class, field capacity (FC), permanent wilting point (PWP), and saturation percentage were estimated using SPAW-hydrology software (*USDA-NRCS, 2013*) based on the soil texture result. Plant available water capacity (AWC) was determined as the difference between FC and PWP (*Estefan, Sommer & Ryan, 2013*). A combined glass electrode was used to measure soil pH and electric conductivity (EC) in a suspension of soil and deionized water (1:2.5 w/v) (*Mclean, 1983*; *Rhoades, 1996*). Soil organic carbon was estimated following the Walkley-Black method (*Estefan, Sommer & Ryan, 2013*). Total N was determined by the micro-Kjeldahl digestion method (*Bremner, 1996*). Available phosphorus (AP) was determined following the Olsen extraction method (*Olsen & Sommers, 1982*). Mehlich 3 extraction was used to determine bioavailable sulfur and boron (*Mehlich, 1984*). Their concentrations in soil digests were measured using Perkin Elmer Optima 8300 Inductively

Coupled Plasma-Optical Emission Spectrometer (ICP-OES). Soil micronutrients (Fe and Zn) were extracted with Diethylene Triamine Penta acetic Acid-triethanolamine (DTPA-TEA) method as described by *Lindsay & Norvell (1978)*, and their concentration in the soil digests was measured using ICP-MS (Inductively Coupled Plasma mass spectroscopy; Perkin-ElmerNexion 300×). Ammonium acetate (NH$_4$OAC, pH-7) leaching (*Ross & Ketterings, 1995*) was used to estimate cation exchange capacity (CEC). Laboratory works were done at Tigray Soil Laboratory Centre, Mekelle (Ethiopia), and Plant Nutrition Laboratory, College of Environmental Science Resources, Zhejiang University, Hangzhou (China).

## Nutrient use efficiency

Nutrient use efficiency (NUE) is a critically important concept in the evaluation of crop production systems. The objective of nutrient use is to increase the overall performance of cropping systems by providing economically optimum nourishment to the crop while minimizing nutrient losses from the field (*Fixen et al., 2015*). Two indicators were used to evaluate nutrient use efficiency (NUE) in this study: (i) Agronomic efficiency of NPSZnB (AE) and (ii) partial factor productivity of NPSZnB (PFP) was calculated (*Fageria & Baligar, 2003*). Agronomic efficiency (AE) and partial factor productivity (PFP) are valuable measures of nutrient use efficiency indices as they provide an integrative index that quantifies total economic output relative to the utilization of all nutrient resources in the system (*Yadav, 2003*). The AE indicates the economic production obtained per unit of NPSZnB blended nutrient applied (*Elias, Teklu & Tefera, 2020*). The PFP, a ratio of the grain yield to the applied nutrient, is a valuable measure of nutrient-use efficiency as it provides an integrative index that quantifies total economic output relative to the utilization of all nutrient resources in the system, including native soil nutrients and nutrients from applied fertilizers (*Dobermann, 2005*; *Yadav, 2003*). They were calculated using the following equations, respectively:

$$AE \left( \frac{kg\ grain}{kg\ NPSZnB} \right) = \frac{G_{NPSZnB} - G_{NP}}{Na} \tag{1}$$

$$PFP \left( \frac{kg\ grain}{kg\ NPSZnB} \right) = \frac{Nn}{Na} \tag{2}$$

where; G$_{NPSZnB}$ = grain yield obtained from plots fertilized with NPSZnB blended fertilizer; G$_{NP}$ = grain yield obtained from plots fertilized with NP fertilizer (urea and DAP); Nn = the total grain yield obtained from each treatment; and Na = the quantity of nutrients applied.

## Partial budget and marginal rate of return analysis

Partial budget analysis was computed using the *CIMMYT (1988)* procedure to determine the economic feasibility of NPSZnB blended fertilizer for optimal bread wheat productivity under rainfed conditions in Ayiba soils. The 10% down adjusted grain and straw yield data were used with local filed prices of 13.5 and 3.5 ETB kg$^{-1}$, respectively. Marginal rate

**Table 3 The physicochemical properties of the experimental fields before sowing in 2017.**

| Physical properties | Soil type | | | Chemical properties | Soil type | | |
|---|---|---|---|---|---|---|---|
| | Vertisol | Cambisol | Mean | | Vertisol | Cambisol | Mean |
| Sand (%) | 22.5 | 26.2 | 24.4 | pH | 7.4 | 7.3 | 7.4 |
| Silt (%) | 29.5 | 17.5 | 23.5 | EC (dSm$^{-1}$) | 0.2 | 0.2 | 0.2 |
| Clay (%) | 48 | 46.5 | 47.3 | SOC (%) | 1.4 | 2.1 | 1.8 |
| Textural class | Clay | Clay | Clay | TN (%) | 0.1 | 0.2 | 0.2 |
| BD (g cm$^{-3}$) | 1.4 | 1.2 | 1.3 | C:N | 8.5 | 11.7 | 10.1 |
| FC (%) | 38.5 | 41.3 | 39.9 | P (mg kg$^{-1}$) | 14.4 | 21.4 | 17.9 |
| PWP (%) | 24.4 | 28.7 | 26.6 | S (mg kg$^{-1}$) | 0.7 | 1.1 | 0.9 |
| AWC (%) | 14.1 | 13.6 | 13.9 | B (mg kg$^{-1}$) | 0.4 | 1.8 | 1.1 |
| Saturation (%) | 49.3 | 49.4 | 49.4 | Zn (mg kg$^{-1}$) | 0.2 | 0.4 | 0.3 |
| – | – | – | – | Mn (mg kg$^{-1}$) | 9.6 | 13.5 | 11.6 |
| – | – | – | – | Fe (mg kg$^{-1}$) | 20.7 | 19.5 | 20.1 |
| – | – | – | – | K$^+$ (cmol$_{(+)}$ kg$^{-1}$) | 0.7 | 0.6 | 0.7 |
| – | – | – | – | CEC (cmol$_{(+)}$ kg$^{-1}$) | 41.8 | 36.9 | 39.4 |

**Note:**
BD, bulk density; FC, field capacity; PWP, permanent wilting point; AWC, available water content; SOC, soil organic carbon; TN, total nitrogen; CEC, cation exchange capacity.

of return analysis (MRR) was executed on non-dominated treatments to pinpoint treatments with the highest return to farmer's investment, considering MRR of 100% is realistic for the recommendation.

## Statistical analysis

The data obtained from lab analysis were checked to test the data sets' normality with the Shapiro-Wilk normality test, and the normality assumption was not violated. All results were reported as means ± standard error (SE) for three replicates. Statistical analysis was conducted with two-way ANOVA to obtain the effect of the model in R software (*R Core Team, 2020*) using package '*doebioresearch*' version 0.1.0 (*Popat & Banakara, 2020*). Whenever significant differences among treatment means have been detected the analysis of variance used the Fisher's least significant difference (LSD) at 5% level in the *doebioresearch package* of the R programming language hosted in R studio. The Pearson's correlation coefficients were calculated between the analyzed agronomic parameters in each soil type using the R '*corrplot package*' (*Wei & Simko, 2021*). All bar graphs were drawn using the OriginPro software (*OriginLab, 2019*).

## RESULT AND DISCUSSION

### Pre-planting soil physicochemical analysis of the experimental fields

The result of pre-planting soil physicochemical analysis of the experimental fields (Table 3) indicated in the Vertisol and Cambisol soil types that the soil textural class was clay and neutral with pH of 7.4 and 7.3, respectively. The bulk density of the studied soils varied from 1.2 to 1.4 g cm$^{-3}$, which was found ideal for plant growth (*Hazelton & Murphy, 2016*). The electrical conductivity (EC) rate in the two soils was low, indicating

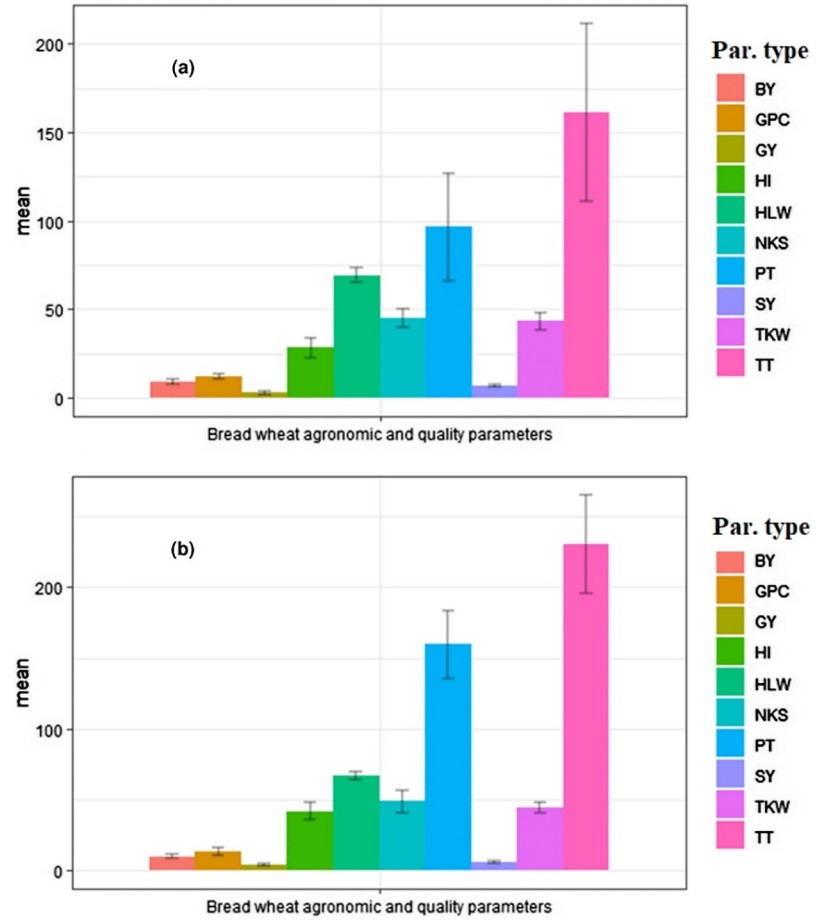

**Figure 1 Bar graph showing the mean ± SD of agronomic and grain quality data measured for (A) Vertisol and (B) Cambisol soils.** The data are pooled of two cropping seasons (2017–2018). Bars indicated that standard deviation (SD). The colors in the legend showed the different parameter types evaluated (BY, biological yield; GPC, grain protein content; GY, grain yield; HI, harvest index; HLW, hectoliter weight; NKS, number of kernels per spike; PT, productive tillers; SY, straw yield; TKW, thousand kernel weight; and TT, total tillers).

non-saline. According to EthioSIS rating (*Karltun et al., 2013*), available Boron (B) was found in a very low rate in Vertisol and optimum rate in Cambisol. Vertisol was found low in SOC and TN and very low in Av Olsen P. Whereas, Cambisol was found optimum in SOC and TN and low in Av Olsen P. Available sulfur (S) was found in very low status in both soils. Both soil types recorded high CEC value (>35 $cmol_{(+)}$ $kg^{-1}$) indicating that all have worthy water holding capacity as described in *Weil & Brady (2017)*, optimum exchangeable $K^+$, high Mn and Fe, and low Zn micronutrients. The rating for most soil chemical properties discussed above was done based on *Elias (2016)*.

## Descriptive and analysis of variance statistics

Figure 1 represents the mean statistical evaluation of the obtained agronomic data and quality trait results. Table 4 summarized the basic descriptive statistics for all agronomic and quality parameters of bread wheat to measure the range of variability. The descriptive

**Table 4 Descriptive statistics of bread wheat agronomic parameters (two seasons pooled data: 2017–2018) grown in different soils.**

| Parameters | Min. | Median | Mean | Max. | SD |
|---|---|---|---|---|---|
| *Haplic Vertisol* | | | | | |
| Total tillers per meter square (TT) | 89.25 | 169.25 | 161.14 | 243.25 | 50.4 |
| productive tillers per meter square (PT) | 53.55 | 101.55 | 96.48 | 145.5 | 30.3 |
| Number of kernels per spike (NKS) | 36.5 | 45.36 | 45.04 | 53.6 | 5.01 |
| Thousands Kernel Weight (g) (TKW) | 35.12 | 43.39 | 43.17 | 51.45 | 4.77 |
| Biological yield (t ha$^{-1}$) (BY) | 6.5 | 8.88 | 9.13 | 12.3 | 1.43 |
| Grain yield (t ha$^{-1}$) (GY) | 1.4 | 2.35 | 2.63 | 4.6 | 0.9 |
| Straw yield (t ha$^{-1}$) (SY) | 5.1 | 6.48 | 6.51 | 7.7 | 0.59 |
| Harvest Index (%) (HI) | 20.25 | 26.52 | 28.04 | 37.4 | 5.41 |
| Grain protein content (%) (GPC) | 9.75 | 12.21 | 12.1 | 14.6 | 1.38 |
| Hectoliter weight (kg hl$^{-1}$) (HLW) | 60.5 | 70.65 | 69.1 | 74.7 | 4.26 |
| *Haplic Cambisol* | | | | | |
| Total tillers per meter square (TT) | 176.3 | 225.5 | 230.7 | 283.5 | 34.8 |
| productive tillers per meter square (PT) | 117.4 | 159.3 | 159.6 | 198.4 | 23.9 |
| Number of kernels per spike (NKS) | 35.3 | 47.75 | 48.47 | 67.4 | 7.95 |
| Thousands Kernel Weight (g) (TKW) | 36.9 | 44.98 | 44.5 | 52.35 | 3.78 |
| Biological yield (t ha$^{-1}$) (BY) | 6.7 | 9.85 | 9.65 | 11.9 | 1.34 |
| Grain yield (t ha$^{-1}$) (GY) | 2.43 | 4.2 | 4.07 | 5.38 | 0.93 |
| Straw yield (t ha$^{-1}$) (SY) | 4.04 | 5.7 | 5.59 | 6.83 | 0.75 |
| Harvest Index (%) (HI) | 29.22 | 42.04 | 41.77 | 51.19 | 6.07 |
| Grain protein content (%) (GPC) | 9.2 | 12.82 | 12.99 | 21.2 | 2.70 |
| Hectoliter weight (kg hl$^{-1}$) (HLW) | 61.5 | 65.7 | 66.49 | 71.6 | 3.04 |

**Note:**
Min., minimum; Max., maximum; SD, standard deviation.

statistics showed considerable variation for all agronomic data between the soil types treated with different NPSZnB blended and blanket NP fertilizers. Accordingly, Cambisol was found superior in tillering capacity and recorded 30.15% and 39.55% higher in total and productive tillers m$^{-2}$ than Vertisol. Likewise, higher biological yield (BY), grain yield (GY), harvest index (HI), and grain protein content (GPC) was recorded in Cambisol. Whereas, in Vertisol soil highest straw yield (SY) and hectoliter weight (HLW) was recorded by 16.46% and 3.93%, respectively, than Cambisol.

Moreover, median values were almost near the mean values, representing the nonappearance of outliers in calculating the central tendency for the agronomic data analysis (Table 4). In Table 5 are summarized values of the mean squares with *p*-values of the effects of soil type (st), fertilizer rate (fr), and their interactions (st*fr) for all traits evaluated for this study. The analysis of variance (Table 5) showed statistically significant variation among agronomic data due to the main factor's effect and their interaction (except straw yield) in Ayiba under rainfed conditions.

**Table 5 Mean square and *p*-value computed for yield and quality traits (two seasons pooled data: 2017–2018).**

| F-test source | DF | Mean squares and significance | | | | | | | | | |
|---|---|---|---|---|---|---|---|---|---|---|---|
| | | TT | PT | NKS | TKW | BY | GY | SY | HI | GPC | HLW |
| Soil type (st) | 1 | 58,015*** | 47,784*** | 141.02*** | 21.3** | 3.32*** | 24.85*** | 10*** | 2,262.41*** | 9.49*** | 81.64*** |
| Fertilizer rate (fr) | 7 | 11,318*** | 4,465*** | 227.64*** | 100.61*** | 10.56*** | 5.22*** | 1.15** | 161.99*** | 24.96*** | 77.81*** |
| st*fr | 7 | 739*** | 247*** | 18.65* | 8.24* | 0.66* | 0.22*** | 0.48ns | 20.3* | 2.79*** | 9.67*** |
| Residuals | 30 | 49 | 35 | 7.64 | 2.74 | 0.24 | 0.01 | 0.26 | 7.06 | 0.57 | 0.59 |

Note:

DF, degree of freedom; TT, total tillers; PT, productive tillers; NKS, number of kernels per spike; TKW, thousand kernels weight; BY, biological yield; GY, grain yield; SY, straw yield; HI, harvest index; GPC, grain protein content; HLW, hectoliter weight; An asterisk (\*), two asterisks (\*\*), and three asterisks (\*\*\*) denotes significant at $p < 0.05$, 0.01, and 0.001, respectively and ns is not significant at $p < 0.05$.

### Response of wheat yield components, yield, and quality attributes

According to the analysis of variance results presented in Table 5 above, the interaction effect of soil types and fertilizer rate significantly ($p < 0.001$) affected TT (m$^{-2}$), PT (m$^{-2}$), NKS (grains spike$^{-1}$), TKW (g), BY (t ha$^{-1}$), GY (t ha$^{-1}$), HI (%), GPC (%), and HLW (k hl$^{-1}$). At the same time, straw yield (t ha$^{-1}$) was not significantly affected by the interaction effect. However, it was significantly affected by the main effect of soil type ($p < 0.001$) and fertilizer rate ($p < 0.01$). The average value of the agronomic parameters and GPC was higher in Cambisol than Vertisol, indicating that Cambisol was better in soil nutrient availability. However, the average HLW was high in Vertisol than Cambisol, indicating that grain harvested from Vertisol brings the best price and provides the best quality, which is more valuable to the end-user. The lower HLW in Cambisol compared to Vertisol may be due to endured stress at some point during the grain-filling period or when frost ends the growing season before physiological maturity, which is a common problem in the study area. The model's analysis showed a highly statistically significant relationship between the measured bread wheat agronomic and quality parameters and the independent factors. Therefore, about 74.3–99.3% ($R^2$) of the agronomic and quality measures variations can be attributable to variations in soil type and fertilizer application. The $R^2$ indicated the variation present in the agronomic and quality parameters explained by the model. Results are presented in Figs. 2–4 and Tables 6–8.

### Total and productive tillers (m$^{-2}$)

Tillering capacity is an essential trait of plant architecture for grain yields, and the number of tillers per plant determines spike number and affects grain production directly (*Naruoka et al., 2011*; *Shang et al., 2021*; *Tesfaye, Laekemariam & Habte, 2021*). In this study, the analysis of variance revealed a highly significant ($p < 0.001$) interaction effect between soil type and fertilizer rate for both total and productive tillers (m$^{-2}$) of wheat (Table 5). The result indicated that the highest number of total and productive tillers (m$^{-2}$) was observed in $T_8$ for Vertisol and Cambisol soils, which was also statistically at parity with $T_5$ and $T_2$ in Vertisol and with the two predecessor treatments in Cambisol (Fig. 2). The increase in the number of tillers in response to an increasing rate of NPSZnB blended fertilizer indicated the importance of soil nutrients other than the NP in the NPSZnB as a limiting factor for better vegetative growth and crop development. Plots

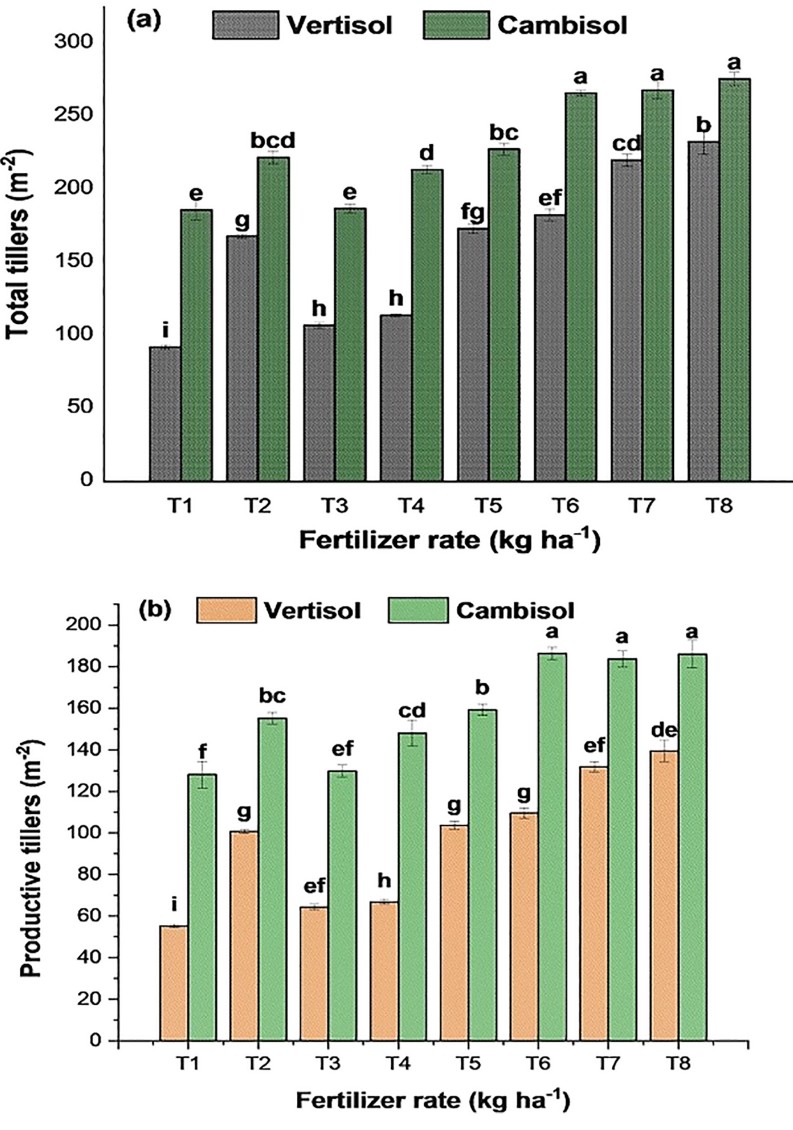

**Figure 2** Interaction effects of soil type and fertilizer rate on (A) total tillers ($m^{-2}$) and (B) productive tillers ($m^{-2}$) of bread wheat (two seasons pooled data: 2017–2018). Values followed by similar letters are not significantly different at $p < 0.05$ according to LSD test. Error bars indicate standard error of the mean.

treated with NPSZnB produced significantly superior wheat total and productive tillers compared to the unfertilized control plots.

The finding of this study revealed that under Ayiba condition, application of NPSZnB blended fertilizer: ≥100 kg ha$^{-1}$ in both soils produced significantly higher total tillers ($m^{-2}$) and productive tillers ($m^{-2}$) than the conventional NP application. Besides, the highest mean total and productive tillers ($m^{-2}$) recorded in Vertisol was 27.8% and 27.7% higher and in Cambisol was 19.5% and 16.6% higher, respectively, compared to the NP blanket application. The highest record in both soils indicated that the application of blended fertilizer has an advantage on tillering capacity of bread wheat over the conventional application under Ayiba conditions. The lower numbers of total and

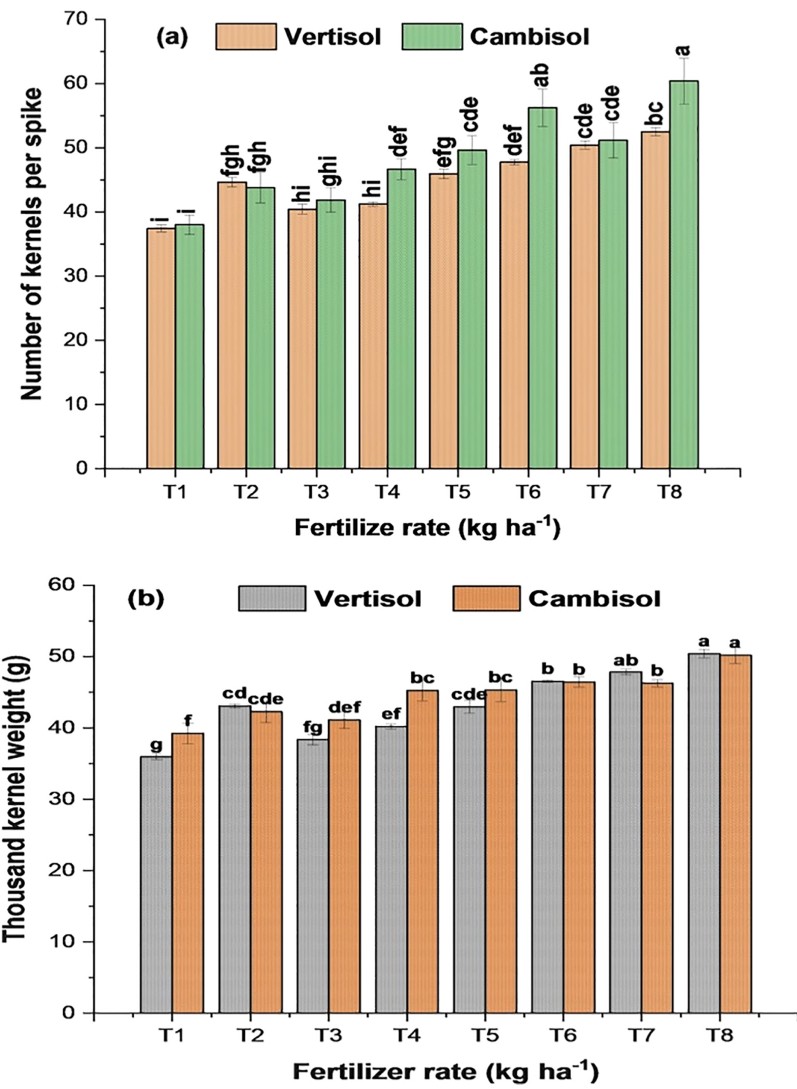

**Figure 3 Interaction effects of soil type and fertilizer rate on (A) number of kernels per spike and (B) thousand grain weight of bread wheat (two seasons pooled data: 2017–2018).** Values followed by similar letters are not significantly different at $p < 0.05$ according to LSD test. Error bars indicate standard error of the mean.                      

productive tillers (m$^{-2}$) were produced from control treatment in both soils, at par with T$_3$ in Cambisol (Fig. 2). The variation in tillering capacity as a function of blended fertilization between soil types proved that the NP blanket recommendation was wrongly used without considering the limitation of other elemental soil nutrients, which play a significant role in plant growth and development. However, the part of NP that stimulates the formation of new tillers and prevents abortion of formed tillers is indispensable. Therefore, soil-specific fertilization practice seems a very meaningful approach in the study area for telling capacity.

The present study result agrees with the findings of *Abera, Tana & Dessalegn (2020).* They reported the highest TT and PT for durum wheat varieties (Mangudo and Utuba) treated with NPSB blended fertilizer in central Ethiopia. Similarly, the highest number of

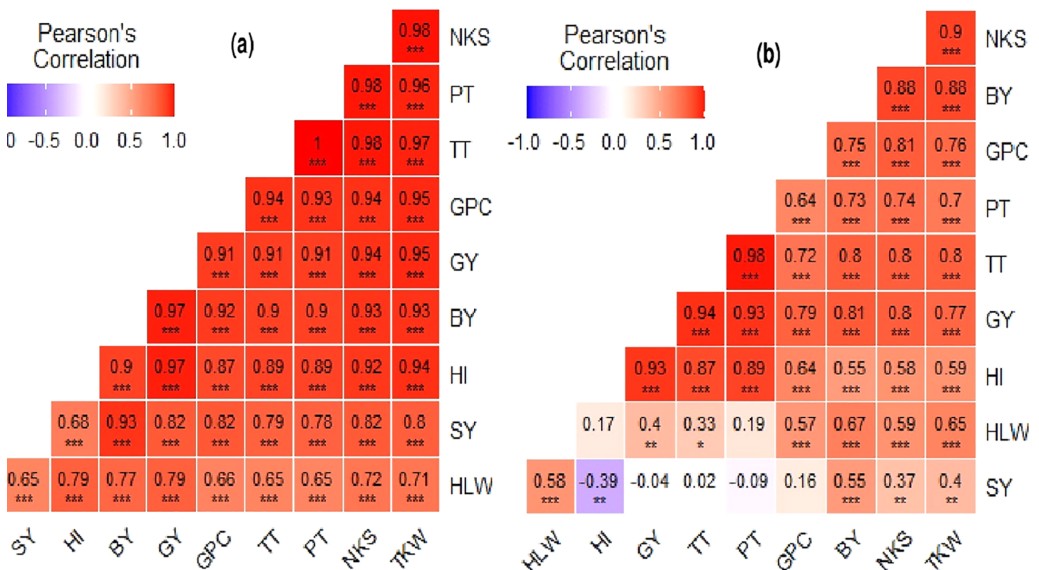

**Figure 4 Pearson's correlation matrix graph for agronomic data of bread wheat grown in (A) Vertisol and (B) Cambisol soils of the Ayiba area, northern Ethiopia.** Color intensity is proportional to the correlation coefficients.

**Table 6 The interaction effect of soil type and fertilizer rates on yield attributes of bread wheat (two season pooled data: 2017–2018).**

| Treatments (kg ha$^{-1}$) | Biological yield (t ha$^{-1}$) | | Grain yield (t ha$^{-1}$) | | Harvest index (%) | |
|---|---|---|---|---|---|---|
| | Vertisol | Cambisol | Vertisol | Cambisol | Vertisol | Cambisol |
| T$_1$ | 7.22 ± 0.36$^i$ | 7.58 ± 0.47$^{hi}$ | 1.58 ± 0.10$^j$ | 2.49 ± 0.04$^h$ | 21.92 ± 0.48$^h$ | 33.23 ± 2.73$^{ef}$ |
| T$_2$ | 8.50 ± 0.23$^{efg}$ | 9.20 ± 0.44$^{defg}$ | 2.13 ± 0.12$^i$ | 4.21 ± 0.04$^{cd}$ | 25.06 ± 0.08$^{gh}$ | 46.11 ± 2.68$^{ab}$ |
| T$_3$ | 8.12 ± 0.13$^{gh}$ | 8.67 ± 0.33$^{efg}$ | 1.77 ± 0.09$^j$ | 2.96 ± 0.02$^g$ | 21.76 ± 0.93$^h$ | 34.22 ± 1.13$^e$ |
| T$_4$ | 8.33 ± 0.22$^{fgh}$ | 8.87 ± 0.55$^{defg}$ | 2.07 ± 0.09$^i$ | 3.76 ± 0.07$^e$ | 24.88 ± 1.59$^{gh}$ | 42.78 ± 3.45$^{bc}$ |
| T$_5$ | 9.13 ± 0.17$^{def}$ | 10.22 ± 0.30$^{bc}$ | 2.63 ± 0.15$^h$ | 4.12 ± 0.08$^d$ | 28.81 ± 1.28$^{fg}$ | 40.37 ± 0.54$^{cd}$ |
| T$_6$ | 9.65 ± 0.08$^{cd}$ | 10.90 ± 0.35$^b$ | 3.23 ± 0.07$^f$ | 4.78 ± 0.08$^b$ | 33.5 ± 0.56$^e$ | 43.95 ± 1.66$^{bc}$ |
| T$_7$ | 10.18 ± 0.07$^{bc}$ | 10.87 ± 0.23$^b$ | 3.27 ± 0.06$^f$ | 4.86 ± 0.02$^b$ | 32.08 ± 0.61$^{ef}$ | 44.74 ± 0.84$^{abc}$ |
| T$_8$ | 11.93 ± 0.20$^a$ | 10.98 ± 0.37$^b$ | 4.33 ± 0.15$^c$ | 5.36 ± 0.02$^a$ | 36.29 ± 0.60$^{fde}$ | 48.86 ± 1.54$^a$ |
| Mean | 9.13 | 9.66 | 2.63 | 4.07 | 28.04 | 41.77 |
| CV | 5.22 | | 3.56 | | 7.61 | |
| LSD$_{5\%}$ | 0.82 | | 0.2 | | 4.43 | |
| R$^2$ (%) | 92.1 | | 99.3 | | 94.4 | |

**Note:**
Results are Mean ± SE, means followed by different letters in a column and row are significantly different according to LSD ($p < 0.05$) test, LSD, least significant difference; CV, coefficient of variation.

tillers per plant of wheat due to the combined application of 200 kg NPS + 92 kg N ha$^{-1}$ was reported by *Seyoum, Jemal & Tamado (2017)*. Other research findings also indicated that the application of NPK fertilizers had a potential role in the number of TT and PT production of wheat (*Abera, Tana & Dessalegn, 2020*; *Baque et al., 2006*; *Getachew & Dechassa, 2014*; *Malghani et al., 2010*) due to their positive role in stimulating vegetative growth and development. In southern Ethiopia, the higher number of TT and PT from the

**Table 7 The main effect of soil type and fertilizer treatments on wheat straw yield.**

| Soil type | Straw yield (t ha$^{-1}$) |
|---|---|
| Vertisol | 5.59 ± 0.39[b] |
| Cambisol | 6.51 ± 0.14[a] |
| LSD$_{5\%}$ | 0.3 |
| Fertilizer treatments (kg ha$^{-1}$) | |
| Control (no fertilizer) | 5.36 ± 0.21[c] |
| 100 DAP+50 KCl +100 Urea | 5.68 ± 0.21[bc] |
| 50 NPSZnB+50 KCl+119.8 Urea | 6.03 ± 0.21[ab] |
| 75 NPSZnB+50 KCl+110 Urea | 5.69 ± 0.21[bc] |
| 100 NPSZnB+50 KCl+100.4 Urea | 6.30 ± 0.21[a] |
| 125 NPSZnB+50 KCl+90.7 Urea | 6.26 ± 0.21[ab] |
| 150 NPSZnB+50 KCl+81.1 Urea | 6.46 ± 0.21[a] |
| 175 NPSZnB+50 KCl+71.3 Urea | 6.61 ± 0.21[a] |
| LSD$_{5\%}$ | 0.61 |
| CV | 8.48 |
| R$^2$ (%) | 74.3 |

Note:
Results are mean ± SE, Values followed by similar letters along column are not significantly different at $p = 0.05$ according to LSD test, LSD, least significant difference; CV, coefficient of variation.

**Table 8 The interaction effect of soil type and fertilizer treatments on bread wheat grain quality parameters (two season pooled data: 2017–2018).**

| Fertilizer rate (kg ha$^{-1}$) | Grain protein content (%) | | Hectoliter weight (kg hl$^{-1}$) | |
|---|---|---|---|---|
| | Vertisol | Cambisol | Vertisol | Cambisol |
| Treatment 1 | 9.93 ± 0.1[g] | 9.38 ± 0.18[g] | 63.92 ± 0.25[g] | 62.23 ± 0.38[h] |
| Treatment 2 | 12.43 ± 0.1[de] | 12.43 ± 0.10[de] | 62.07 ± 0.85[h] | 65.00 ± 0.79[fg] |
| Treatment 3 | 10.58 ± 0.13[fg] | 10.17 ± 0.27[g] | 66.53 ± 0.19[e] | 63.90 ± 0.29[g] |
| Treatment 4 | 11.56 ± 0.1[ef] | 12.08 ± 0.16[de] | 69.95 ± 0.20[d] | 65.30 ± 0.26[ef] |
| Treatment 5 | 11.93 ± 0.12[ef] | 13.28 ± 0.16[cd] | 71.28 ± 0.09[c] | 65.75 ± 0.83[ef] |
| Treatment 6 | 12.67 ± 0.2[de] | 14.58 ± 0.21[b] | 71.95 ± 0.35[bc] | 68.98 ± 0.34[d] |
| Treatment 7 | 13.25 ± 0.2[cd] | 14.00 ± 0.10[bc] | 72.83 ± 0.14[b] | 69.45 ± 0.22[d] |
| Treatment 8 | 14.40 ± 0.1[bc] | 17.95 ± 1.63[a] | 74.25 ± 0.43[a] | 71.30 ± 0.17[c] |
| Mean | 12.09 | 12.98 | 69.10 | 66.49 |
| CV | 6 | | 1.14 | |
| LSD$_{5\%}$ | 1.26 | | 1.28 | |
| R$^2$ (%) | 92.3 | | 97.5 | |

Note:
Results are Mean ± SE, means followed by different letters in a column and row are significantly different according to LSD ($p < 0.05$) test, LSD, least significant difference; CV, coefficient of variation.

combined effect of NPS and KCl was reported than the unfertilized plots (*Tesfaye, Laekemariam & Habte, 2021*). Similarly, the highest number of TT (421 tillers m$^{-2}$) and PT (375.7 tillers m$^{-2}$) were reported with the application of NPSB supplied with N in central highland Ethiopia (*Desta & Almayehu, 2020*). *Leghari et al. (2016)* and

*Abayu (2012)* reported a significant and highest tillering in wheat and Teff with NPKB and NPSZnMg blended fertilizers. In agreement with our findings, *Jan, Amanullah & Noor (2011)* also reported significantly higher PT m$^{-2}$ with the application of 30 Mg FYM ha$^{-1}$ + 90 kg N ha$^{-1}$. The PT per hill of rice was significantly higher in integrated nutrient management than chemical fertilizer alone in the Indian Agricultural Research Institute, New Delhi, India (*Singh, Singh & Sharma, 2013*).

### Number of kernels per spike and thousand kernels weight

The number of kernels per spike (NKS) and thousand kernel weight (TKW) are important yield contributing parameters and directly affect wheat grain yield. The 2-year average data showed that NKS varied from 37.42 to 52.48 grains per spike in Vertisol and 38.03 to 60.38 grains per spike in Cambisol. Likewise, TKW varied from ~36 to 50.4 g in Vertisol and 39.2 to 50.2 g in Cambisol. The highest mean NKS and TKW were recorded with the application of 175 kg NPSZnB ha$^{-1}$ in both Vertisol (52.5 grains per spike, 50.4 g) and Cambisol (60.4 grains per spike, 50.2 g) soils (Fig. 3). At the same time, the minimum NKS and lighter TKW were recorded in the control treatment in both soils. In general, the mean value of NKS and TKW has displayed an increasing trend as the application rate of blended fertilizer increased (Fig. 3) in both soils, reflecting the importance of NPSZnB in wheat NKS and TKW.

In this field experiment, bread wheat grown on fertilized plots showed 7.4–28.7% and 6.3–28.7% higher in NKS and TKW over the control treatments at Vertisol and Cambisol soils, respectively. The highest NKS and TKW may be due to adequate and better nutrition of the plants resulting in good grain filling and development of better seed size. As reported in *Usman, Tamado & Wogeyehu (2020)* it could also be due to the provision of balanced nutrients, which enhanced spike elongation and accumulation of assimilating in the grains and thus resulted in more spikelets per spike and heavier grains. Explicitly, the TKW is an essential indication of flour yield where wheat can be classified according to its grain weight as 15–25 g (very small), 26–35 g (small), 36–45 g (medium), 46–55 g (large) and over 55 g (very large) (*Williams et al., 1986*). Accordingly, TKW obtained in this study fall under medium to large size in both soils. The larger seed size was produced by applying >125 kg NPSZnB ha$^{-1}$ in both soils. According to the overall mean, Cambisol produced higher TKW by ~3% than Vertisol. Compared to NP recommendation, application of blended fertilizer >100 kg ha$^{-1}$ and >125 kg ha$^{-1}$ in Vertisol produced significantly higher NKS and TKW, respectively. Whereas, the application of NPSZnB >75 kg ha$^{-1}$ produced significantly higher NKS and TKW in Cambisol soil (Fig. 3). The variation in NKS and TKW in both soil types as a function of fertilization indicated that different soils have various responses to different fertilizer rates and types.

In agreement with the present study, *Desta & Almayehu (2020)* reported the highest (50.07) NKS under combined application of 150/92 kg NPSB/N ha$^{-1}$ fertilizer rates. Increasing NKS was also reported by *Malghani et al. (2010)* and *Tesfaye, Laekemariam & Habte (2021)* due to the increasing rate of NPKS fertilizer. *Debnath et al. (2011)* and *Muhammad et al. (2009)* also reported that Boron application significantly affected wheat's

NKS. Similarly, a substantial difference with the application of fertilizers blended by macro/micronutrient nutrients which significantly increased TKW of teff was also reported by *Fayera, Adugna & Mohammed (2014)*. The present result also agrees with the finding of *Yasir et al. (2015)* in Pakistan. Recent studies on NPS and K (*Tesfaye, Laekemariam & Habte, 2021*) and NPKSZn (*Brhane, Mamo & Teka, 2017*) demonstrated that adding these nutrients together improved photosynthetic activity and enhanced sink transport of the grain and resulted in heavier grains. In the sub-humid environment, *Mubeen et al. (2021)* found a similar result as the highest NKS and heavier TKW was recorded in plots where the integrated application of natural and synthetic sources at equal dose was used. However, *Liu, Liao & Liu (2021)* revealed that excessive nitrogen fertilizer use and high planting density reduce kernel number per spike and TKW in wheat.

### Biological yield (BY), grain yield (GY), straw yield, and harvest index (HI)

The result revealed that mean BY varied from 7.2 to 11.9 t ha$^{-1}$ (at Vertisol) and from 7.6 to 11 t ha$^{-1}$ (at Cambisol), and mean GY varied from 1.6 to 4.3 t ha$^{-1}$ (at Vertisol) and 2.5 to 5.4 t ha$^{-1}$ (at Cambisol). The result also indicated that application of NPSZnB blended >100 kg ha$^{-1}$ in Vertisol produced higher BY (6.9–28.8%), higher GY (19–50.8%), and higher HI (13–30.9%) compared to the application of NP alone. Likewise, in Cambisol soil, higher BY (~10–16.2%), GY (11.9–21.5%), and HI (5.9%) were recorded with the application of >100, >125, and 175 kg NPSZnB ha$^{-1}$, respectively compared to the application of NP alone. Plots treated with NPSZnB gave 10.7–63.5% and 15.9–53.5% higher GY in Vertisol and Cambisol compared to the control plot, respectively. The highest yield could be attributed to the relatively balanced nutrients in NPSZnB, resulting in enhanced yield due to better nutrient use efficiency and the synergistic effect of nutrients in the new compound fertilizer.

The highest and lowest BY and GY were produced in both soil types treated with the highest rate and lowest rate (control plots), respectively (Table 6). The highest BY and GY attained from the highest blended fertilizer rate than the positive control NP is due to the presence of S, Zn, and B minerals which play a vital role: in metabolic processes, synergistic utilization of other nutrients by plants, enzyme activation, enhancement of photosynthesis and assimilate transport processes from source to sink during the growth period. We also observed that the average BY, GY, and HI increased in most, as both soil types increased the NPSZnB blended fertilizer rate applications. The result enumerated that BY, GY, and HI of wheat responded to varying application rates of NPSZnB blended fertilizer in the experimental soils, indicating that soil-specific fertilization is an important approach. Bread wheat grain yields in most plots receiving NPSZnB blend and NP fertilizers in both soil types were significantly greater than the unfertilized control plots (Table 6).

Although several previous research findings (*Gessesew, Mohammed & Woldetsadik, 2015*; *Rurinda et al., 2020*; *Vanlauwe et al., 2015*) reported that increasing NP fertilizers stimulate growth and development and increase the uptake of other nutrients from the soil. However, their sole application trend over a long time brings adverse side effects on the soil nutrient stock, environmental and health concerns, cost and availability of the

exclusive mineral fertilizers unless transformed to new fertilizer formulations (*Agegnehu, Vanbeek & Bird, 2014*; *Bindraban et al., 2012*; *Elias, 2018*; *Mugwe, Ngetich & Otieno, 2019*; *Vanlauwe et al., 2015*). Other studies also indicated that micronutrients are used to increase crop productivity, especially when conventional NPK fertilizers are not efficient (*Dimkpa & Bindraban, 2016*). The present results regarding yield attributes follow the findings of others (*Abera et al., 2020b*; *Desta & Almayehu, 2020*; *Tola et al., 2020*), who reported that the highest wheat and maize grain yield was attained from the application of blended fertilizer at Ambo and Toke Kuyaye districts of Ethiopia. Similarly, a recent study also reported higher wheat and maize grain yield response for soil test-based fertilizer recommendations than the NP blanket application alone (*Elias, Teklu & Tefera, 2020*; *Rurinda et al., 2020*). Earlier studies have reported that combined application of mineral and organic fertilizer resulted in synergistic effects and improved synchronization of nutrient release and uptake by plants leading to higher grain yield (*Abdou et al., 2016*; *Jan, Amanullah & Noor, 2011*; *Rezig, Elhadi & Mubarak, 2013*; *Saha et al., 2008*).

Our result is also in accord with other findings (*Desta & Almayehu, 2020*; *Hřivna, Kotková & Burešová, 2015*; *Muhammad et al., 2009*; *Tola et al., 2020*), who stated that the application of fertilizers blended with micronutrients enhanced the vegetative growth of bread wheat and ultimately increased biomass production due to sufficient assimilation process as a result of synergistically balanced nutrient supply. In southern Ethiopia, *Elka & Laekemariam (2020)* reported increased BY of haricot beans with an increase in the rate of NPS and organic fertilizers. The result is also in agreement with the research findings of others (*Astatke et al., 2004*; *Singh & Wanjari, 2014*; *Tesfaye, Laekemariam & Habte, 2021*) reported in different soil types. As *Amanullah & Inamullah (2016)* reported, applying P + Zn increases totaled dry matter accumulation and partitioned more significant amounts into the reproductive plant parts (panicles), resulting in a higher harvest index. In accord with the present finding, significantly higher HI results were attained from the application of blended fertilizers, as reported by *Dejene & Chala (2021)* and *Fisseha et al. (2020)* in Ethiopia. Besides, significant variations were detected in HI of winter wheat in Southern Bavaria, Germany (*Sticksel et al., 2000*). *Mubeen et al. (2021)* reported that integrated application of natural and synthetic sources each @ 60 kg ha$^{-1}$ produced maximum GY in a sub-humid environment. Increased HI in rice was also reported with combined application of P and Zn than sole application (*Amanullah & Inamullah, 2016*; *Mafi, Sadeghi & Doroodian, 2013*). Significantly higher GY was obtained with the combined application of S and P than their sole application (*Assefa, Haile & Tena, 2021*). However, contrary to the present result, *Amare et al. (2019)* reported that K, Zn, and B did not bring a significant difference in BY compared to the blanket application, and *Lemma & Tana (2015)* and *Desta & Almayehu (2020)* also reported as the application of blended fertilizer has no significant effect on HI of wheat.

Regarding the straw yield (SY), the two-season pooled data revealed that it was significantly affected by the main effect of soil type and fertilizer treatments (Table 5). The straw yield produced from Cambisol was higher by 14.1% over Vertisols irrespective of fertilizer effect treatment, which could be due to better nutrient availability in Cambisols than Vertisols for bread wheat growth (Table 7). Regarding the effect of fertilizer

treatment, the higher SY was recorded with application 175 kg ha$^{-1}$ which was at par with treatment 3, 5, 6, and 7; whereas, the lowest was recorded at control which was statistically similar with treatment 1, 2, and 4 (Table 7). Compared to the conventional NP recommendation (T$_2$), the application of NPSZnB >150 kg ha$^{-1}$ produced higher SY (Table 7). The advantage on SY attribute by NPSZnB blended fertilizer was probably caused by greater availability and uptake of macro/micronutrients that might have resulted in higher photosynthesis, tissue differentiation, and translocation assimilation turn, leading to better vegetative growth.

Previous studies reported that blended fertilizer was found vital to increase straw yield. The present study result is in line with the finding of *Tekle & Wassie (2018)* and *Jafer (2018)*. They reported that the SY of teff and maize was found highest in blended fertilizers compared to the NP blanket recommendation. The positive response of NPS fertilizer to achieve food security in Ethiopia was also reported by *Tamene et al. (2017)*. The present finding was also in line with that of *Tesfay & Gebresamuel (2016)*. They reported that SY of teff was significantly affected by the application of blended fertilizer and exceeded 7% and 490% over the recommended NP and control plots, respectively. Others (*Mubshar et al., 2012*; *Soni, Swarup & Singh, 1996*; *Ullah et al., 2018*) also found that SY of wheat and rice increased significantly with increasing Mn and B application rates. Straw yields of *Tef* were also reported significantly increased due to the application of NPSB on Vertisols of Hatsebo, central Tigray (*Tewolde, Gebreyohannes & Abrha, 2020*), and application of K in Vertisols of central highland Ethiopia (*Demïss et al., 2020*). The current finding by *Lakshmi et al. (2021)* revealed a significantly higher SY where wheat was treated with 10 kg Zn ha$^{-1}$ in calcareous soil. Application of Sulfur on Vertisol and Cambisol soil types reported improving straw yield in central highland Ethiopia (*Assefa, Shewangizaw & Kassie, 2020*).

### Grain protein content (GPC) and hectoliter weight (HLW)

The growing population has increased the need for wheat-based products as a result of which a focus on the end-use quality is very much essential. The quality of wheat is largely based on the wheat storage proteins which extensively influence the dough properties (*Branković et al., 2018*; *Sharma et al., 2020*). Increasing grain protein and its strength has also recently received greater attention due to its positive effect on bread, pasta, and noodle products (*Cato & Mullan, 2020*; *Goel et al., 2021*; *Johansson, Prieto-Linde & Svensson, 2004*; *Kinyua et al., 2006*). Because, the protein content in grain (flour) is the main quality criterion, especially for bread-producing wheat (*García-Molina & Barro, 2018*) which determines flour's water-absorbing ability, stability, resistance, and elasticity. In this study, increased GPC with increased NPSZnB blende fertilizer quantity in both soils is observed. In Vertisols and Cambisols, GPC ranged from 9.93% to 14.4% and 9.38% to 17.95%, respectively, of which all are almost within the acceptable range, which proves the response of wheat to NPSZnB and NP fertilization. Both soils recorded the highest and lowest GPC in T$_8$ (175 kg ha$^{-1}$) and T$_1$ (controls). The mean GPC for overall treatments was 12.1% and ~13% in Vertisols and Cambisols, respectively (Table 8). The result indicated that the increased application of NPSZnB led to a subsequent increase in total GPC,

thereby suggesting that an increase in multi-nutrient (mainly N and S: the significant constituents of protein) availability in the rhizosphere ultimately increases the rate of uptake, translocation, assimilation in leaves and reassimilation into developing grains.

According to the ISO-20483 method of test (https://www.iso.org/standard/59162.html) treatment 7 and 8 in Vertisols and treatment 5 to 8 in Cambisols produced grade 1 standard grain, which can be considered the best quality for baking. The other treatments in both soils made grade 2 to 4 standard grain, except control treatments produced below grade 4 standards. This indicates that wheat production under Ayiba conditions requires soil-specific fertilization to have quality grain that fills the maximum standard limit set by ISO-20483 for protein content. The plausible reasons for variation in GPC between soils (locations) could be attributed to available soil N and S contents. According to *Couch et al. (2017)*, of the N absorbed by the plant, 31–60% remobilizes to capsules and seeds, that why nitrogen is a ubiquitous nutrient in the environment. Sulfur is also an essential component of amino acids (like cysteine and methionine) crucial to protein formation and improves cereal crops' milling and baking quality (*Clarkson & Hanson, 1980*). Zinc is also engaged in more than 300 enzymes for protein and carbohydrate metabolism with a significant constitute for human immunity enhancement (*Lakshmi et al., 2021*). Studies conducted so far to analyze the effect of N fertilizer on grain quality traits have also revealed significant increases in total GPC of rice, wheat, maize, and barley under an increased rate of N application (*Carson & Edwards, 2009*; *Chandel et al., 2010*; *Nishizawa, 2005*).

More importantly, research findings indicated that the deposition of protein in grains depends on the plethora of interconnected metabolic pathways involved in the uptake of N, S, Zn, and other elements with synergistic effects from the soil, their transport to source tissues such as leaves and mobilization and remobilization to developing grains (*Chandel et al., 2010*; *Grusak, 2002*). The present study results were similar to many other reports, which explained that the GPC of wheat increased with increasing N fertilizer rates (*Bereket et al., 2014*; *Dargie, Wogi & Kidanu, 2020*; *Fisseha et al., 2020*). Similarly, *Abera, Tana & Dessalegn (2020)* reported the highest GPC from the highest NPSB blended fertilizer application. The highest crude protein content was recorded in southwestern Ethiopia with the application of 200 kg NPSB $ha^{-1}$ + 46 kg P $ha^{-1}$ + 128 kg N $ha^{-1}$ as reported by *Zewide, Singh & Kassa (2021)*. The results obtained in this study also substantiate with *Tao et al. (2018)*, who noted that sulfur fertilization increased grain and protein yields, grain weight, and total starch.

Regarding the hectoliter weight (HLW), it is a general physical indicator of grain quality in all wheat grading systems (*Brennan, Samaan & El-Khayat, 2012*; *Dexter & Marchylo, 2000*). Higher HLW usually means higher quality grain; therefore, more valuable to the end-user. In this study, the highest HLW (74.3 kg $hl^{-1}$) was recorded from the highest NPSZnB rate (175 kg $ha^{-1}$) for Vertisol, while the lowest HLW (62.2 kg $hl^{-1}$) was recorded from the control treatment for Cambisol (Table 8). Both soils showed significant differences in their HLW, but it was found as Vertisol > Cambisol based on the average result. The result showed 66.7% of the NPSZnB blend fertilizer treatment at Vertisol and 16.7% at Cambisol produced wheat grain with HLW above 70 kg $hl^{-1}$ (Table 8). According

**Table 9 Effect of NPSZnB blended fertilizer rate on AE and PFP of bread wheat grown under two soil types (two seasons pooled data: 2017–2018).**

| Treatments (kg ha$^{-1}$) | TNA | Vertisol | | | Cambisol | | |
|---|---|---|---|---|---|---|---|
| | | GY | AE | PFP | GY | AE | PFP |
| Treatment 1 | 0 | 1.58 | – | – | 2.49 | – | – |
| Treatment 2 | 135 | 2.13 | – | 15.78 | 4.21 | – | 31.19 |
| Treatment 3 | 112 | 1.77 | −3.21 | 15.80 | 2.96 | −11.16 | 26.43 |
| Treatment 4 | 123.4 | 2.07 | −0.49 | 16.77 | 3.76 | −0.36 | 30.47 |
| Treatment 5 | 134.7 | 2.63 | 3.71 | 19.52 | 4.12 | −0.67 | 30.59 |
| Treatment 6 | 146.1 | 3.23 | 7.53 | 22.11 | 4.78 | 3.90 | 32.72 |
| Treatment 7 | 157.7 | 3.27 | 7.23 | 20.74 | 4.86 | 4.12 | 30.82 |
| Treatment 8 | 169.1 | 4.33 | 13.01 | 25.61 | 5.36 | 6.80 | 31.70 |

Note:
TNA, total nutrient applied (according to Table 2); GY, grain yield; AE, agronomic efficiency; PFP, partial factor productivity.

to the ES ISO-7971/2 method of the test (*ES, 2017*) for the wheat standard of quality (kg hl$^{-1}$), treatment 8 in Vertisols produced grade 3 standard grain. Treatment 5 to 7 in Vertisol and treatment 8 in Cambisol also produced grade 4 standard grain. The other treatments in both soils were below grade 4-grain quality standards. The higher HLW with the application of the highest NPSZnB fertilizer might be due to the role of balanced nutrients on wheat quality, such as flour yield and protein content. This indicates that wheat production under Ayiba conditions requires more soil-specific balanced fertilization than the highest setting to produce quality grains that fill the maximum standard limit set by ES ISO-7971/2 for hectoliter weight. *Abera, Tana & Dessalegn (2020)* reported the highest HLW (80.2 kg hl$^{-1}$) with the application of 183 kg NPSB ha$^{-1}$. Others also reported HLW of 78.5–83.4 kg hl$^{-1}$ for durum wheat varieties in Ethiopia (*Fana et al., 2012*; *Woyema, Bultosa & Taa, 2012*). Our result is also close to the findings of other researchers (*Muhammad et al., 2009*; *Seyoume, 2006*; *Soboka, Bultossa & Eticha, 2017*) who reported HLW variations from 68.3 to 82.5 kg hl$^{-1}$ in different Pakistan and Ethiopian wheat varieties. The variation in hectoliter weight reported differs probably due to varieties, soil type, climate, and agronomic practices.

## Nutrient use efficiency indices

There has been poor synchrony between crop nutrient demand and nutrient supply because of a limited understanding of the nutrient uptake-yield relationship. Hence, developing an integrated soil-crop system management strategy that simultaneously increases grain yield and nutrient use efficiency (NUE) is required. In this study, the nutrient use indices of agronomic efficiency (AE) and partial factor productivity (PFP) for bread wheat were found to vary among treatments and soil types in the Ayiba condition. The two season's average grain yield, AE, and PFP data exhibit variation among the soil types (Table 9). In Vertisol and Cambisol soils, bread wheat treated with 175 kg NPSZnB ha$^{-1}$ gave maximum AE of 13.01 and 6.8 kg grains kg$^{-1}$ NPSZnB, respectively. The lowest AE results (negative) were recorded at the NPSZnB rate of 50–75 kg ha$^{-1}$

(for Vertisol), and rate 50–100 kg ha$^{-1}$ (for Cambisol) compared to the blanket application rate (Table 9).

This implies that application of NPSZnB blended <75 kg ha$^{-1}$ (in Vertisol) and <100 kg ha$^{-1}$ (in Cambisol) have little agronomic value than the blanket application in Ayiba condition. As *Tamene et al. (2017)* explained, fertilizer efficiency should be improved by applying a balanced and appropriate fertilizer mix, increasing crop yield, improving soil health, and increasing the revenue from fertilizer application. Thus, application of NPSZnB blended >100 kg ha$^{-1}$ (in Vertisol) and >125 kg ha$^{-1}$ (in Cambisol) showed better agronomic value on wheat than the NP alone in Ayiba condition. Similarly, *Puniya et al. (2019)* reported that higher uptake of Fe, Mn, Cu, and Zn were obtained with combined FYM and NPK application compared to mineral sources of NPK alone. Now, emphasis is also placed on improving the use efficiency of fertilizers through the 4R nutrient stewardship principle ( *i.e.*, right source, right rate, right time, and right placement) (*IPNI, 2014*). *Singh, Singh & Sharma (2013)* also found the seed quality parameters like germination rate and vigor indexes and N uptake and soil organic carbon content were higher in integrated nutrient management than chemical fertilizer alone. Another study by *Chandel et al. (2010)* confirms that soil containing balanced nutrient composition in the rhizosphere enhances the uptake, translocation, and redistribution of nutrients into grains.

On the other hand, *Gupta & Khosla (2012)* and *Ruisi et al. (2015)* reported that the crop was not effectively utilizing 50–60% of applied N fertilizer. Hence, increasing N use efficiency in cereal cropping systems by just 10% could result in an annual savings of US$5 billion and substantial improvement in environmental quality (*Gupta & Khosla, 2012*). Given the regional differences, it is also feasible to primarily identify a regional soil nutrient status and then adjust according to the actual site conditions to increase NUE. In this field study, the observed negative average AE in both soil types indicated that NPSZnB blended fertilizer has no advantage compared to the conventional application. However, it is evident that N fertilization is crucial in increasing grain productivity and quality; but, to achieve maximum profitable production and minimize negative environmental impact, improving N use efficiency by applying other limited micronutrients together should be considered.

Regarding PFP, the average PFP showed a positive relationship with the NPSZnB blended fertilizer rate (Table 8). Maximum PFP of 25.61 (38.4% higher than $T_2$) and 32.72 (4.7% higher compared to $T_2$) kg grains kg$^{-1}$ NPSZnB blend was detected when bread wheat was treated with 175 kg NPSZnB ha$^{-1}$ in Vertisol and with 125 kg NPSZnB ha$^{-1}$ in Cambisol, respectively. Bread wheat treated with NPSZnB blended treatments provided higher PFP than NP fertilizer in Vertisol soil while varied in Cambisol soil (Table 8). This indicates that in Vertisol soil type, bread wheat was positively responding to NPSZnB blended fertilizer rates.

## Correlations among agronomic and grain quality parameters

The correlation analysis was determined to observe the degree of relationship among agronomic and grain quality traits (Fig. 4). Correlation coefficients among most of the

**Table 10 Partial budget and marginal rate of return analysis of bread wheat productivity by NPSZnB blended fertilizer rate experiment for Ayiba Vertisol and Cambisol soils (two season pooled data: 2017–2018).**

| soil type | FR (kg ha$^{-1}$) | AGY (kg ha$^{-1}$) | ASY (kg ha$^{-1}$) | FC (Birr ha$^{-1}$) | SC (Birr ha$^{-1}$) | TVC (Birr ha$^{-1}$) | MC | GFBG (Birr ha$^{-1}$) | GFBS (Birr ha$^{-1}$) | TGFB (Birr ha$^{-1}$) | NB = TGFB-TVC | MNBC | MRR (%) | Rank |
|---|---|---|---|---|---|---|---|---|---|---|---|---|---|---|
| | 0 | 1,422 | 3,717 | 0 | 1,440 | 1,440 | – | 19,197 | 13,009.5 | 32,206.5 | 30,766.5 | – | – | – |
| | 50 | 1,917 | 4,023 | 2,257.1 | 1,440 | 3,697.1 | 2,257.1 | 25,879.5 | 14,080.5 | 39,960 | 36,262.9 | 5,496.4 | 243.5 | 6 |
| | 75 | 1,593 | 4,527 | 2,549.3 | 1,440 | 3,989.3 | 292.2 | 21,505.5 | 15,844.5 | 37,350 | 33,360.8 | D | D | – |
| Vertisol | NP | 1,863 | 5,337 | 2,750 | 1,440 | 4,190 | 200.8 | 25,150.5 | 18,679.5 | 43,830 | 39,640 | 6,279.3 | 3,127.9 | 3 |
| | 100 | 2,367 | 5,220 | 2,843.8 | 1,440 | 4,283.8 | 93.8 | 31,954.5 | 18,270 | 50,224.5 | 45,940.7 | 6,300.7 | 6,717.2 | 1 |
| | 125 | 2,907 | 5,778 | 3,137.2 | 1,440 | 4,577.2 | 293.4 | 39,244.5 | 20,223 | 59,467.5 | 54,890.4 | 8,949.7 | 3,050.8 | 4 |
| | 150 | 2,943 | 6,363 | 3,431.7 | 1,440 | 4,871.7 | 294.6 | 39,730.5 | 22,270.5 | 62,001 | 57,129.3 | 2,239 | 760.1 | 5 |
| | 175 | 3,897 | 6,903 | 3,723.9 | 1,440 | 5,163.9 | 292.2 | 52,609.5 | 24,160.5 | 76,770 | 71,606.2 | 16,715.8 | 5,721.7 | 2 |
| | 0 | 2,241 | 4,995 | 0 | 1,440 | 1,440 | – | 30,253.5 | 17,482.5 | 47,736 | 46,296 | – | – | – |
| | 50 | 3,789 | 4,788 | 2,257.1 | 1,440 | 3,697.1 | 2,257.1 | 51,151.5 | 16,758 | 67,909.5 | 64,212.4 | 17,916.4 | 793.9 | 4 |
| | 75 | 2,664 | 5,031 | 2,549.3 | 1,440 | 3,989.3 | 292.2 | 35,964 | 17,608.5 | 53,572.5 | 49,583.3 | D | D | – |
| Cambisol | NP | 3,384 | 5,220 | 2,750 | 1,440 | 4,190 | 200.8 | 45,684 | 18,270 | 63,954 | 59,764 | D | D | – |
| | 100 | 3,708 | 5,481 | 2,843.8 | 1,440 | 4,283.8 | 93.8 | 50,058 | 19,183.5 | 69,241.5 | 64,957.7 | 745.3 | 794.7 | 3 |
| | 125 | 4,302 | 5,490 | 3,137.2 | 1,440 | 4,577.2 | 293.4 | 58,077 | 19,215 | 77,292 | 72,714.9 | 7,757.2 | 2,644.3 | 1 |
| | 150 | 4,374 | 5,292 | 3,431.7 | 1,440 | 4,871.7 | 294.6 | 59,049 | 18,522 | 77,571 | 72,699.3 | D | D | – |
| | 175 | 4,824 | 5,625 | 3,723.9 | 1,440 | 5,163.9 | 292.2 | 65,124 | 19,687.5 | 84,811.5 | 79,647.7 | 6,932.8 | 2,373 | 2 |

**Note:**
FR, Fertilizer Rate; AGY, adjusted Grain Yield; ASY, Adjusted straw Yield; FC, Fertilizer cost; SC, seed cost; TVC, total variable cost; MC, marginal cost; TGFB, total gross field benefit; NB, net benefit; ETB, Ethiopian Birr; MNB, marginal net benefit; MRR, marginal rate of return; D, dominated (**N.B:** DAP and NPSZnB = 15.5 ETB kg$^{-1}$, Urea = 12 ETB kg$^{-1}$, wheat grain = 13.5 ETB kg$^{-1}$, wheat straw = 3.5 ETB kg$^{-1}$).

characteristics were statistically significant and revealed a noticeable association among the parameters measured in both soil types. The correlation analysis among the agronomic and grain quality parameters is presented in Fig. 4A for Vertisol and in Fig. 4B for Cambisol soils, respectively. Specifically, the correlation results: at Vertisol farm among parameters were positive and significant (Fig. 4A), and at Cambisol soil, most parameters exhibited considerable positive correlation, except SY with HI revealed negative and significant correlation (Fig. 4B). Correspondingly others (*Getachew & Dechassa, 2014*; *Tesfay & Gebresamuel, 2016*; *White & Wilson, 2006*) also reported similar results on teff and wheat.

## Partial budget and marginal analysis

Partial budget and marginal rate of return analysis of NPSZnB blended fertilizer rate and blanket recommendation NP fertilizer across the studied two soil types are summarized in Table 10. For partial budget economic analysis, the grain and straw yields were reduced by 10% to reflect the difference between the experimental yield and the yield farmers could expect from the same treatment. Accordingly, all treatments produced a higher positive net benefit (NB) than the control in both soil types. In both Vertisol and Cambisol soils, the highest net benefit of 45,940.7 ETB ha$^{-1}$ and 72,714.9 ETB ha$^{-1}$ with MRR of 6,717.2% and 2,644.3% was obtained from the application of 100 kg ha$^{-1}$ and

125 kg ha$^{-1}$ NPSZnB blended fertilizer, respectively (Table 10). The result suggests that for every 1 ETB invested in fertilizer in Vertisol and Cambisol, farmers (producers) can expect to recover the 1 ETB and obtain an additional 45.9 and 72.7 ETB, respectively.

The MRR results are ranked, and the three highest consecutive alternatives are highlighted in boldface for optional recommendation (Table 10). Therefore, the application of blended fertilizer: 100 kg NPSZnB ha$^{-1}$ in Vertisol and 125 kg NPSZnB ha$^{-1}$ in Cambisol soils are recommended for farmers in the Ayiba area. This study further proved that the application of NP alone was not profitable but can be an optional alternative in Vertisol soils. The highest net benefit in response to applying NPSZnB blended fertilizer could be attributed to incorporating additional soil nutrients rather than NP alone; thereby, the productivity of bread wheat increased. Hence, implementing the recommended soil-specific blended fertilizer in the Ayiba area would create a pathway to wheat self-sufficiency. Moreover, as *Elias, Okoth & Smaling (2019)* noted, such investment in the national wheat sector would create more job opportunities in agricultural value chains.

## CONCLUSION

The manifestation of multi-nutrient insufficiency in Ethiopian soils is backed by nutrient depletion and disproportionate fertilizer application. For the past five decades, the country has relied only on urea and DAP fertilizers to boost soil production. However, the farming life did not escape the poverty cycle throughout the half-century that thrived on these fertilizers. The main causes of failure were (i) imbalanced input, (ii) neglecting the status of other important plant nutrients in the fertilization strategy since they were thought to be sufficient, and (iii) soil deterioration. The status of other critical plant nutrients was found to be in reverse, and new soil-specific fertilizers combined with the macro/micronutrients in a region were introduced to boost soil and crop production. To get larger yields per unit of fertilizer applied, there is also a requirement to improve fertilizer usage efficiency. In light of this, a field experiment was conducted in Ayiba, northern Ethiopia, in 2017 and 2018 cropping seasons to study the response of bread wheat to the application of NPSZnB blended fertilizer rates on two distinct soil types (Vertisol and Cambisol) under rainfed conditions.

The impact of soil type and fertilizer rate on wheat production and quality parameters was investigated in this study. According to the findings of this study, using NPSZnB mixed fertilizer in both soil types is a crucial fertilizing strategy for boosting bread wheat yield, yield component, and quality attributes. The findings of the two-season combined study demonstrated that the application rate of NPSZnB blended fertilizer had a significant ($p < 0.001$) impact on TT, PT, NKS, TKW, BY, GY, HI, GPC, and HLW in both soils (Table 5). Bread wheat grain yields were greater in Cambisol soil than in Vertisol soil. In this study, NPSZnB mixed fertilizer application increased bread wheat production and yield components in all soils in Ayiba under rainfed conditions. As a consequence, in both soil types, using a high NPSZnB mixed fertilizer led to the maximum production of bread wheat. This suggested that the soil's natural nutrient concentration was insufficient to manage production. Therefore, the finding of this research showed

producers to consider soil-specific fertilization rather than recommending blanket from one site success to another which was practiced in the last more than 50 years and failed to secure the food production in Africa and Ethiopia.

The average AE and PFP statistics from the two seasons show that there is variance among the soil types. Bread wheat treated with 175 kg NPSZnB ha$^{-1}$ exhibited the highest AE of 13.01 and 6.8 kg grains kg$^{-1}$ NPSZnB, respectively, in Vertisol and Cambisol soils. On the other hand, the observed negative average agronomic nutrient efficiency (AE) in both soil types indicated that applying the NPSZnB blended fertilizer has no advantage compared to the conventional Urea and DAP fertilizers. When bread wheat was treated with 175 kg NPSZnB ha$^{-1}$ in Vertisol and 125 kg NPSZnB ha$^{-1}$ in Cambisol, respectively, a maximum PFP of 25.61 (38.4% higher than $T_2$) and 32.72 (4.7% higher than $T_2$) kg grains kg$^{-1}$ NPSZnB blend were discovered. This research reveals that for wheat production in the study site, using a soil-specific fertilization technique is preferable. As a result, treatments of 100 and 125 kg NPSZnB ha$^{-1}$ on Vertisol and Cambisol soils, respectively, were efficacious and economically viable under Ayiba conditions. These rates gave the highest marginal rate of return, according to a partial budget study (6717.2% and 2644.3%, respectively). Furthermore, testing at a greater rate and detecting nutrient synergism are two clear management strategies that need more research to boost wheat production and fertilizer usage efficiency.

### Funding
The Mekelle University (Ethiopia)—CASCAPE project provided PhD funding for this study. The funders had no role in study design, data collection and analysis, decision to publish, or preparation of the manuscript.

### Grant Disclosures
The following grant information was disclosed by the authors:
The Mekelle University (Ethiopia).

### Competing Interests
The authors declare that they have no competing interests.

### Author Contributions
- Weldemariam Seifu conceived and designed the experiments, performed the experiments, analyzed the data, prepared figures and/or tables, authored or reviewed drafts of the paper, and approved the final draft.
- Eyasu Elias conceived and designed the experiments, performed the experiments, analyzed the data, prepared figures and/or tables, authored or reviewed drafts of the paper, and approved the final draft.
- Girmay Gebresamuel conceived and designed the experiments, performed the experiments, analyzed the data, prepared figures and/or tables, authored or reviewed drafts of the paper, and approved the final draft.

- Wolde Tefera conceived and designed the experiments, performed the experiments, analyzed the data, prepared figures and/or tables, authored or reviewed drafts of the paper, and approved the final draft.

## Data Availability

The raw data are available in the Supplemental File.

## Supplemental Information

Supplemental information for this article can be found online at http://dx.doi.org/10.7717/peerj.13344#supplemental-information.

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
