# Peer review of "Soil type and fertilizer rate affect wheat (Triticum aestivum L.) yield, quality and nutrient use efficiency in Ayiba, northern Ethiopia"

_PeerJ, doi:10.7717/peerj.13344_

## Round 0.1 · original submission · Major Revisions

Reviewers' comments on your work have now been received. The manuscript has been assessed by three reviewers. Reviewers indicated that the abstract, the introduction, the method, the result and discussion section should be improved. Moreover, English editing and academic writing service is needed. I agree with this evaluation and I would, therefore, request for the manuscript to be revised accordingly.

Reviewers have requested that you cite specific references. You may add them if you believe they are especially relevant. However, I do not expect you to include these citations, and if you do not include them, this will not influence my decision.

·

Basic reporting

General Comment
The research study is an extensive investigation that covers a much-needed area of knowledge. It commendably seeks to solve the issue of fertilizer use and application with regards to soil fertility and the quality of wheat grown in a part of Ethiopia.
However, several issues should be revised before the manuscript gets published.

Experimental design

Scope and concept

In the abstract, the authors stated that “Bread wheat (Triticum aestivum) is the most important staple crop…” This claim is seemingly untrue and should be completely removed. Though the region where this claim was intended for was not mentioned, the claim is invalid for world and for Ethiopia. Global rankings of the most important staple foods place Maize Corn and Rice above Wheat (https://www.worldatlas.com/articles/most-important-staple-foods-in-the-world.html). In Ethiopia, maize was reportedly more produced in the 90’s and 2000’s than wheat [Rashid, Shahidur. (2010). Staple Food Prices in Ethiopia. Michigan State University, Department of Agricultural, Food, and Resource Economics, Food Security Collaborative Working Papers.] Even the authors listed (in order) sorghum, teff, and maize ahead of wheat for the study region (line 153). Contrastingly, the authors mentioned in lines 44- 46 “Bread wheat (Triticum aestivum) is ONE OF THE MAJOR CEREALS, along with maize (Zea mays), teff (Eragrostis teff Zucc), barley (Hordeum vulgare) and sorghum (Sorghum bicolor).” WHERE? IN THE WORLD? OR IN ETHIOPIA?

In lines 52-54, authors stated “However, wheat yields are WOEFULLY LOW, with a national average of 2.97 t ha-1 under farmers conditions”. How is “woefully low” defined? The word woeful should be removed

The study was conducted in a region of the Northern part of Ethiopia. In lines 140-141, it was only stated that “The area is among the potential wheat-producing regions in Tigray furnished with tepid to cool semi-arid agro-ecological zone”. The authors are yet to establish a good justification to use a research study conducted in that one region of the country to conclude for the whole country (lines 663-665). I think a more brief but effective justification should be included after line 141.

Validity of the findings

Lengthy manuscript
The manuscript is seemingly too wordy. Every section contains too much information; some of which are not required, and the required ones missing. The introduction covers 3 pages. The materials and methods cover 4. Results and discussions take around 13.5 pages. The conclusion consumes 1.5 pages with 3 massive paragraphs. This should be totally reviewed. The entire manuscript should be rid-off of information and data that is excessive or do not drive any point. An example is for lines 153-157 which speaks “extremely extensively” on the too-many crops (13) grown in the study region.

Discussions and unanswered questions.

The results and discussions section are summarily a declaration of the results obtained and results obtained in other published studies. Detailed descriptions that answer the question of “WHY?” where hardly mentioned. This should be revised in the entire discussion section.
For example, lines 323-324 states “The increase in the number of tillers in response to an increasing rate of NPSZnB blended fertilizer indicated the importance of balanced nutrients for better vegetative growth and crop development. In addition, all fertilized treatments significantly improved wheat total and productive tillers compared to the unfertilized control.” The question is WHY? Why the increase? What caused it? How did the fertilizers improve wheat production? Did they regulate the soil pH? Which micronutrients did they supply? Did they affect any soil circulation?

Same concerns are for the following lines amongst several others:
Line 332: “…the application of blended fertilizer has an advantage on 333 tillering capacity of bread wheat over the conventional application under Ayiba conditions…” HOW? WHY?
Lines 376-378: “The highest NKS and TKW may be due to the provision of balanced nutrients, which enhanced spike elongation and accumulation of assimilating in the grains and thus resulting in more spikelets per spike and heavier grains.” This looks like a technical discussion. It should be expatiated with reference to quality literature.
Line 387: “…The variation in NKS and TKW in both soil types as a function of fertilization indicated that different soils have various responses to different fertilizer rates and types.” HOW? WHY?
Lines 501-502: “Increasing grain protein and its strength has recently received greater attention due to their positive effect on bread and pasta products” HOW? WHY?

Additional comments

Grammatical disconnect and incomplete sentences

The introduction and several parts of the manuscript have twisted or unlinked sentences. These sections should be entirely revised for technicality and grammatical soundness and meaningfulness. Few are mentioned below:

Line 54: “…reasons for the low levels of wheat, yields are declining soil” The comma after wheat should be removed
Line 63-64: “In addition, at present, population rising and dietary pattern shifting linked to urbanization are causing to surpasses the demand for national wheat supply (6.3 Mt demand Vs. 4.6 Mt supply)”. This sentence should be re-written for clarity
Line 67: “… for the last several years” Should be removed and re-written with proper grammatical expression.
Line 68: “Although demand and production are not related yet, there is…” HOW?
Line 72: “…2023 (MoA, 2019). Thus, increasing domestic wheat…” The period should be removed. The latter sentence cannot stand without the former
Line 79: “The problem with fertilization indicated that the need for great…” “that” should be removed.
Line 80: “By doing so it is possible to increase the nutrient use…” A comma should be added after “so”

In addition, the use of professional titles for the cover page abstract "Prof." and "Dr.", except required by the journal is apparently unethical. Reviewers can view such as a case for winning favor or bias.

·

Basic reporting

I think this manuscript is interested and provide good findings. Unlike developed countries, Sub Saharan Africa lacks literatures and specialized systems which regularly monitor changes in nutrient budget. Therefore, it is an important contribution to the filed. Therefore, I suggest accepting the manuscript after considering my following suggestions and comments.
1. The English language should be improved to ensure that an international audience can clearly understand your text. For example, the title which is the most important sentence for the manuscript has a mistake (affect NOT affects). I strongly suggest the author to do a very careful proofreading for the entire manuscript. There are so many problems in writing that it is difficult to continue reading the paper. I suggest you have a colleague who is proficient in English and familiar with the subject matter review your manuscript, or contact a professional editing service.
2. The introduction is good and contains good information that paves the way for reaching the idea of the research. But this is a manuscript and not a doctoral thesis. I strongly suggest the authors to reduce this introduction to the half and only information that confirms the idea of the research. I also greatly missed the discussion of the recent findings of Elrys et al. (2019, 2020, 2021) which dealt with the nitrogen budget in Africa during the past five decades, as well as calculating the nitrogen fertilizer rate needed to achieve self-sufficiency of Africa by 2050. Please indicate their important results in the introduction and discuss their accuracy in the discussion section based on some of your results. You can find their papers through the following references:
a. 10. Elrys, A. S., M. K. Abdel-Fattah, S. Raza, Z. Chen, J. Zhou (2019). Spatial trends in the budget of nitrogen flows in the African agro-food system over the past five decades. Environmental Research Letters, 14: 124091.
b. Elrys, A. S., Metwally, M. S., Raza, S., Alnaimy, M. A., Shaheen, S. M., Chen, Z., & Zhou, J. (2020). How much nitrogen does Africa need to feed itself by 2050? Journal of Environmental Management, 268.
c. Elrys, A.S., E.-S.M. Desoky, M.A. Alnaimy, H. Zhang, J.-b. Zhang, Z.-c. Cai,…, Cheng, Y. (2021). The food nitrogen footprint for African countries under fertilized and unfertilized farms. Journal of Environmental Management, 279: 111599.
3. Results and Discussion: I highly recommend the authors to separate this section into two sections. I mean the authors must introduce the results in a separate section and Discussion in another section. Regarding results, there is no need to present each attribute in a separate title. Please group them into groups and show readers your important findings to help them understand the message of your study.
For the discussion section, you can separate it into two subsections, the first section should discuss your important findings but please avoid repeating the results again. The second part of the discussion should be the implications of this study, you have good implications but the authors have to discuss them and not just mention it. Don't forget to link your repercussions with the large scale of Africa because, you know, nutrient depletion is not only a problem for Ethiopia, but for most of sub-Saharan Africa.

Experimental design

I think this manuscript is interested and provide good findings. Unlike developed countries, Sub Saharan Africa lacks literatures and specialized systems which regularly monitor changes in nutrient budget. Therefore, it is an important contribution to the filed. Therefore, I suggest accepting the manuscript after considering my following suggestions and comments.
1. The English language should be improved to ensure that an international audience can clearly understand your text. For example, the title which is the most important sentence for the manuscript has a mistake (affect NOT affects). I strongly suggest the author to do a very careful proofreading for the entire manuscript. There are so many problems in writing that it is difficult to continue reading the paper. I suggest you have a colleague who is proficient in English and familiar with the subject matter review your manuscript, or contact a professional editing service.
2. The introduction is good and contains good information that paves the way for reaching the idea of the research. But this is a manuscript and not a doctoral thesis. I strongly suggest the authors to reduce this introduction to the half and only information that confirms the idea of the research. I also greatly missed the discussion of the recent findings of Elrys et al. (2019, 2020, 2021) which dealt with the nitrogen budget in Africa during the past five decades, as well as calculating the nitrogen fertilizer rate needed to achieve self-sufficiency of Africa by 2050. Please indicate their important results in the introduction and discuss their accuracy in the discussion section based on some of your results. You can find their papers through the following references:
a. 10. Elrys, A. S., M. K. Abdel-Fattah, S. Raza, Z. Chen, J. Zhou (2019). Spatial trends in the budget of nitrogen flows in the African agro-food system over the past five decades. Environmental Research Letters, 14: 124091.
b. Elrys, A. S., Metwally, M. S., Raza, S., Alnaimy, M. A., Shaheen, S. M., Chen, Z., & Zhou, J. (2020). How much nitrogen does Africa need to feed itself by 2050? Journal of Environmental Management, 268.
c. Elrys, A.S., E.-S.M. Desoky, M.A. Alnaimy, H. Zhang, J.-b. Zhang, Z.-c. Cai,…, Cheng, Y. (2021). The food nitrogen footprint for African countries under fertilized and unfertilized farms. Journal of Environmental Management, 279: 111599.
3. Results and Discussion: I highly recommend the authors to separate this section into two sections. I mean the authors must introduce the results in a separate section and Discussion in another section. Regarding results, there is no need to present each attribute in a separate title. Please group them into groups and show readers your important findings to help them understand the message of your study.
For the discussion section, you can separate it into two subsections, the first section should discuss your important findings but please avoid repeating the results again. The second part of the discussion should be the implications of this study, you have good implications but the authors have to discuss them and not just mention it. Don't forget to link your repercussions with the large scale of Africa because, you know, nutrient depletion is not only a problem for Ethiopia, but for most of sub-Saharan Africa.

Validity of the findings

I think this manuscript is interested and provide good findings. Unlike developed countries, Sub Saharan Africa lacks literatures and specialized systems which regularly monitor changes in nutrient budget. Therefore, it is an important contribution to the filed. Therefore, I suggest accepting the manuscript after considering my following suggestions and comments.
1. The English language should be improved to ensure that an international audience can clearly understand your text. For example, the title which is the most important sentence for the manuscript has a mistake (affect NOT affects). I strongly suggest the author to do a very careful proofreading for the entire manuscript. There are so many problems in writing that it is difficult to continue reading the paper. I suggest you have a colleague who is proficient in English and familiar with the subject matter review your manuscript, or contact a professional editing service.
2. The introduction is good and contains good information that paves the way for reaching the idea of the research. But this is a manuscript and not a doctoral thesis. I strongly suggest the authors to reduce this introduction to the half and only information that confirms the idea of the research. I also greatly missed the discussion of the recent findings of Elrys et al. (2019, 2020, 2021) which dealt with the nitrogen budget in Africa during the past five decades, as well as calculating the nitrogen fertilizer rate needed to achieve self-sufficiency of Africa by 2050. Please indicate their important results in the introduction and discuss their accuracy in the discussion section based on some of your results. You can find their papers through the following references:
a. 10. Elrys, A. S., M. K. Abdel-Fattah, S. Raza, Z. Chen, J. Zhou (2019). Spatial trends in the budget of nitrogen flows in the African agro-food system over the past five decades. Environmental Research Letters, 14: 124091.
b. Elrys, A. S., Metwally, M. S., Raza, S., Alnaimy, M. A., Shaheen, S. M., Chen, Z., & Zhou, J. (2020). How much nitrogen does Africa need to feed itself by 2050? Journal of Environmental Management, 268.
c. Elrys, A.S., E.-S.M. Desoky, M.A. Alnaimy, H. Zhang, J.-b. Zhang, Z.-c. Cai,…, Cheng, Y. (2021). The food nitrogen footprint for African countries under fertilized and unfertilized farms. Journal of Environmental Management, 279: 111599.
3. Results and Discussion: I highly recommend the authors to separate this section into two sections. I mean the authors must introduce the results in a separate section and Discussion in another section. Regarding results, there is no need to present each attribute in a separate title. Please group them into groups and show readers your important findings to help them understand the message of your study.
For the discussion section, you can separate it into two subsections, the first section should discuss your important findings but please avoid repeating the results again. The second part of the discussion should be the implications of this study, you have good implications but the authors have to discuss them and not just mention it. Don't forget to link your repercussions with the large scale of Africa because, you know, nutrient depletion is not only a problem for Ethiopia, but for most of sub-Saharan Africa.

Reviewer 3 ·

Basic reporting

This is a very well written manuscript; a pleasure to read. This unique, innovative and creative study interestingly examined the effect of NPSZnB blended fertilizer on bread wheat yield, quality traits and use 19 efficiency in two soil types under rainfed conditions in Ayiba, northern Ethiopia.

Experimental design

The focus is well maintained throughout the manuscript. The methods and analyses are well done and results logically presented. Just few comments on Figure 1 : Author can improve this figure and include grids and other elements of a complete map.

Validity of the findings

The results and discussions of this study are well elabolated and are aligned with the previous studies. Conclusions are well stated, linked to original research question and limited to supporting result.

Annotated reviews are not available for download in order to protect the identity of reviewers who chose to remain anonymous.

---

## Round 0.2 · Minor Revisions

Reviewers' comments on your work have now been received. The manuscript has been assessed by three reviewers. Reviewers indicated that the abstract is still too lengthy. Moreover, it is suggested to provide a graphical abstract in the method section. I agree with this evaluation and I would, therefore, request for the manuscript to be revised accordingly.

·

Basic reporting

The manuscript has been greatly improved, but there are some minor concerns that can be considered for a more effective technical delivery of the research message.

The abstract still looks seemingly too lengthy. Several sentences therein would fit in the introduction and discussion sections. The authors should consider minimizing the abstract. The technical abstract should be a crisp but concise summary of the whole research in the fewest words possible.

Experimental design

The design looks better. However, a graphical abstract that summarizes the whole study methodology in one simplified figure would be suggested since it will be highly beneficial to the technical audience perusing this research.

Validity of the findings

The results presented are seemingly satisfactory.

·

Basic reporting

The manuscript has been greatly improved and is recommended for acceptance. Thanks to the authors.

Experimental design

It is OK

Validity of the findings

It is OK

Additional comments

No additional comments

Reviewer 3 ·

Basic reporting

This is a very well written manuscript; it is a pleasure to read. This unique, innovative, and creative study interestingly examined the effects of NPSZnB blended fertilizer on bread wheat yield, quality traits, and use 19 efficiency in two soil types under rainfed conditions in Ayiba, northern Ethiopia.

Experimental design

The focus is well maintained throughout the manuscript. The methods and analyses are well done, and the results are logically presented. The authors have improved the manuscript.

Validity of the findings

The results and discussions of this study are well elabolated and aligned with the previous studies. Conclusions are well stated, linked to the original research question and limited to supporting results.

Annotated reviews are not available for download in order to protect the identity of reviewers who chose to remain anonymous.

---

## Round 0.3 · accepted · Accept

The authors have improved their manuscript.